# Nonpertubative Many-Body Theory for the Two-Dimensional Hubbard Model at Low Temperature: From Weak to Strong Coupling Regimes

Ruitao Xiao[1*], Yingze Su[1*], Junnian Xiong[1], Hui Li[2], Huaqing Huang[1†] and Dingping Li[1‡]

**1** School of Physics, Peking University, Beijing 100871, China
**2** School of Physics, Zhejiang University, Hangzhou 310058, China
⋆ These authors contributed equally to this work.
† huanghq07@pku.edu.cn,        ‡ lidp@pku.edu.cn

November 5, 2025

## Abstract

In theoretical studies of two-dimensional (2D) systems, the Mermin-Wagner theorem prevents continuous symmetry breaking at any finite temperature, thus forbidding a Landau phase transition at a critical temperature $T_c$. The difficulty arises when many-body theoretical studies predict a Landau phase transition at finite temperatures, which contradicts the Mermin-Wagner theorem and is termed a pseudo phase transition. To tackle this problem, we systematically develop a symmetrization scheme, defined as averaging physical quantities over all symmetry-breaking states, thus ensuring that it preserves the Mermin-Wagner theorem. We apply the symmetrization scheme to the GW-covariance calculation for the 2D repulsive Hubbard model at half-filling in the intermediate-to-strong coupling regime and at low temperatures, obtaining the one-body Green's function and spin-spin correlation function, and benchmark them against Determinant Quantum Monte Carlo (DQMC) with good agreement. The spin-spin correlation functions are approached within the covariance theory, a general method for calculating two-body correlation functions from a one-particle starting point, such as the GW formalism used here, which ensures the preservation of the fundamental fluctuation-dissipation relation (FDR) and Ward-Takahashi identities (WTI). With the FDR and WTI satisfied, we conjecture that the $\chi$-sum rule, a fundamental relation from the Pauli exclusion principle, can be used to probe the reliability of many-body methods, and demonstrate this by comparing the GW-covariance and mean-field-covariance approaches. This work provides a novel framework to investigate the strong-coupling and doped regime of the 2D Hubbard model, which is believed to be applicable to real high-$T_c$ cuprate superconductors.

# 1  Introduction

The Hubbard model is considered the simplest model of interacting fermions on a lattice, but it is very difficult to solve or to obtain reasonable approximations across its entire parameter space (for example, temperature and doping). There is sufficient evidence that a full under-standing of it is crucial for exploring unconventional superconductivity (particularly in the con-text of high-$T_c$ superconductivity), antiferromagnetism, the pseudogap phase, strange metal behavior, charge-density waves (CDWs), and many other phenomena. Substantial advances have been made in recent years in investigating the Hubbard model across various regions of its phase diagram, due to the development of different analytic and computational meth-ods [1–5]. The exact location of phase boundaries, for example, the superconducting transition temperature of cuprate superconductors, is still an extremely challenging task [2, 6–9].

## 1.1  Fundamental Relations

For any many-body theory, it is important to preserve three fundamental relations (or iden-tities), while some recently developed powerful computational methods, in particular, dy-namical mean-field theory (DMFT) and its diagrammatic extensions, struggle to satisfy all

of them simultaneously [3]. The first fundamental relation is the fluctuation-dissipation relation (FDR), which connects the response function (i.e., the susceptibility) to the spectral density of fluctuations or the two-body correlators in the frequency domain. The Kubo formula is the time-domain representation of the FDR [10]. The FDR is an exact relation derived by Kubo [11–14]. However, it is not always guaranteed to hold in approximate theories. In any approximation, there is no justification for violating the FDR, and any plausible theory should preserve this relation.

The second fundamental relations are the Ward identities (WIs). For a many-body system with a continuous symmetry, these identities represent rigorous relations between different correlation functions, such as a relation between the one- and two-body Green's functions or a recast relation between the vertex functions and the inverse Green functions. In particular, the Ward-Takahashi identities (WTIs) reflect the conservation of currents [10, 15]. From the WTIs, one can derive the $f$-sum rules, which are fundamentally important for the consistency checking of various theoretical methods and the experimental analysis. The $f$-sum rules are identities that powerfully constrain the dynamics of a many-body system. They state that the total integrals over frequency of the imaginary part of response functions (such as the conductivity) are equal to simple physical properties of its constituents, most commonly the electron mass and density. If the theoretical results violate the $f$-sum rules, the model is considered to be inherently flawed. The $f$-sum rules are widely used to analyze experimental data through tracking the spectral weight transfer in correlated systems and measuring the effective mass and plasma frequency. The WTIs ensure the validity of the $f$-sum rules, thus it is fundamental to maintain them for any theoretical approximation.

The third fundamental relation is the relation due to the local moment sum rules, known as the $\chi$-sum rule [3, 16–19]. This sum rule is given by the relation of some two-body correlators to the one-body correlator at a local site, and is different from other sum rule like the $f$-sum rules which can be derived from the WTIs due to conservation laws [10, 15]. The local moment sum rule ensures that the square of the occupation number of a fixed spin at a fixed site is equal to the occupation number itself, due to the Pauli exclusion principle of the fermionic electron operator. In methods like DMFT and its diagrammatic extensions [3], the impurity solver (calculating the local properties of a single site) must obey this local moment sum rule, and this obedience is a test of the solver's accuracy. Refs. [16–19] present self-consistent theories formulated on the $\chi$-sum rule, which they emphasize.

## 1.2 Methods and Difficulties at Low Temperatures

Hence, to ensure theoretical consistency, it is better to seek an approximate theory that preserves both the WTIs (which reflect conservation laws and yield the corresponding $f$-sum rules) and the $\chi$-sum rule (the local moment sum rule) due to the Pauli exclusion principle. However, it has been conjectured that only the exact solution can meet this standard [3], and finding an approximation to fulfill all requirements appears to be impossible. Therefore, with the exception of exact numerical solutions or exactly solvable theories, most methods will not satisfy all these relations. In the Hartree-Fock (HF) approximation, for instance, the susceptibilities are given by the RPA formulation, where the WTIs (conservation laws due to symmetries) are guaranteed but the $\chi$-sum rule is badly violated in strongly correlated systems (see the detailed calculations in the paper). Below we review some exact numerical calculations which satisfy all those relations.

Determinant Quantum Monte Carlo (DQMC) is an exact numerical method that can satisfy the required relations and capture full spatial correlations, unlike some approaches such as DMFT. Therefore, it is a powerful tool for exploring strongly correlated fermionic systems. However, its application is constrained by computational scaling and the serious fermion sign problem, which appears in the most interesting physical regime: when the system is doped

and at low temperature around the superconducting phase transition temperature [20–24].

To avoid the sign problem in DQMC, Diagrammatic Monte Carlo (DiagMC) was developed as a numerical technique that computes physical quantities by sampling Feynman diagrams [25, 26]. It is indeed notable for its ability to avoid the sign problem. The diagrammatic expansion of the DiagMC is a perturbative series around the HF solutions, thus all identities should be valid. Another advantage of DiagMC is that the calculation can be done directly in the thermodynamic limit, thus avoiding the finite-size scaling analysis necessary for DQMC.

In the doped three-dimensional (3D) Hubbard model, DiagMC was used to study second-order phase transitions by investigating the non-analyticity of physical observables at the transition line from the high-temperature disordered to the low-temperature ordered phase [27]. At low temperature and half filling, DiagMC was also used to determine the antiferromagnetic (AF) critical temperature for couplings up to $U = 6$ (intermediate strength) [28]. For $U > 6$, there is increased difficulty in summing the perturbation series and a loss of accuracy in the critical region near the phase transition. Whether starting from the high-temperature disordered phase [27] or the low-temperature ordered AF phase [28], the critical temperatures are determined only up to $U = 6$. It was suggested in Refs. [27, 28] that DiagMC can capture the Slater-like branch for small $U$, the starting point of the perturbative diagrammatic expansion up to intermediate coupling $U \simeq 6$. $U = 6$ is not in the strongly coupled region for 3D Hubbard model, since $U/W = 0.5$ when $U = 6$, where $W$ is the bandwidth and equals 12 in this model. Generally, it is believed that DiagMC can capture the physics of the weak-coupling branch (the Slater-like branch) due to its perturbative nature but fails to describe the strong-coupling branch (the Mott-Heisenberg branch) [27, 28], as the method is based on sampling perturbative expansions around the mean-field solution.

However, for two-dimensional (2D) many-body systems with continuous symmetry, DiagMC faces an additional challenge. DiagMC was applied to the disordered phase at an inverse temperature of $\beta = 8$ in the 2D Hubbard model [29]. It was found that at filling $n = 0.875$, there is a convergence radius of $R = 5.1$, indicating a Kosterlitz-Thouless phase transition at $U = -5.1$ to 2D superconducting phase. This is consistent with the phase diagram for the attractive Hubbard model presented in Ref. [30]. Ref. [3] indicated that there might be an additional singularity at $U = 6$, which allows us to speculate that solutions at lower temperature (or for strong positive coupling $U$) belong to a branch different from the disordered high-temperature (or for weak positive coupling $U$) branch. For typical cuprate superconductors, the Hubbard $U$ is usually larger than 8 (in units of the hopping energy). The inverse transition temperature, even at optimal doping, is quite high, with $\beta > 30$ reaching values as large as $\beta = 100$ for cuprates with lower transition temperatures at optimal doping. For DiagMC in 2D, we are not aware of any calculations at such low temperatures for $U = 8$, and thus it appears that investigating the low-temperature regime of the 2D Hubbard model is very difficult. Similar difficulties exist even in 2D statistical field theories, from whose research history we can draw some lessons.

For the statistical 2D $XY$ model or the many-body 2D negative $U$ Hubbard model (away from half filling), which is an O(2) invariant microscopic model belonging to the O(2) universality class (the same as that of the 2D $XY$ model) in the corresponding statistical field theory, there is a phase transition known as the Berezinskii-Kosterlitz-Thouless (BKT) transition. This transition involves the binding and unbinding of topological defects, such as vortices in the 2D system, as temperature increases from the low-temperature to the high-temperature regime. Indeed, for the 2D O(2) invariant model, there are two branches: a low-temperature branch with quasi-long-range order where correlations decay algebraically with distance, and a high-temperature branch where correlations decay exponentially. Simulating these O(2) invariant models is particularly challenging due to the very large correlation length in the low-temperature region. Conventional Monte Carlo (MC) simulations must use sufficiently large

lattice sizes to determine whether long-range or quasi-long-range order exists in the thermo-dynamic limit. For large lattice systems, this may require enormous computational resources, which could be beyond the reach of current computational facilities if using MC simulation methods. Fortunately, a tensor network method based on higher-order singular value decomposition has been used to study the 2D $XY$ lattice model [31]. The advantage of this method is that it can evaluate thermal quantities in the nearly infinite lattice size limit and avoids the inherent errors associated with extrapolations from finite-size calculations. This is particularly important for studying the 2D $XY$ model and other 2D spin models in the low-temperature regime.

There has been considerable research effort devoted to the 2D Heisenberg model (an O(3) symmetric lattice model). However, whether a phase transition exists remains an open question. The contradictory results regarding the presence of a phase transition in the 2D Heisenberg model stem from the extremely large correlation length at low temperatures. Recent studies using tensor network methods [32–34] and some Monte Carlo simulations [35, 36] suggest that there is most probably no finite-temperature phase transition in the 2D Heisenberg model, supporting the scenario of asymptotic freedom proposed for the continuous model with O(3) symmetry. However, high-temperature expansions imply that the high-temperature phase is distinct from the low-temperature phase [37,38]. In summary, even if the O(3) symmetric model in 2D does not exhibit a finite-temperature phase transition, there appear to be two different branches: a low-temperature branch and a high-temperature branch.

However, until now, no calculations using Diagrammatic Monte Carlo (DiagMC) applied to the very low-temperature regime (for example, $\beta > 20$, $U = 8$ at half-filling) for the 2D Hubbard model have been reported. DiagMC is a perturbative expansion around the HF solution, which at very low temperatures and half-filling corresponds to the AF solution. The reason might be that, for many-body perturbation theory applied to 2D O($N$) invariant microscopic models with $N > 1$ (such as the 2D Hubbard model, which is O(3) invariant), the perturbative expansion around the symmetry-breaking mean-field solution at low temperatures may suffer from infrared divergences. This behavior is analogous to situation encountered in 2D O($N$) invariant linear or non-linear $\sigma$ models in statistical field theory at low temperatures. In the low-temperature regime (of statistical field theory), the mean-field theory corresponds to a non-zero saddle point of the action, which is a symmetry-broken solution that gives rise to massless Goldstone modes in models with continuous symmetry. However, in 2D, these massless Goldstone modes induce infrared divergences in the perturbative expansion around the mean-field saddle point. As a result, the broken phase is destroyed by these massless excitations, and symmetry restoration occurs. This is the famous Mermin-Wagner theorem [39], which states that continuous symmetries cannot be spontaneously broken in 2D systems at non-zero temperature when interactions are short-ranged.

### 1.3 Lessons from Statistical Field Theory

To tackle the problem of infrared divergences and calculate physical quantities at low temperatures for a statistical field theory with continuous symmetry, a crucial insight was provided by Jevicki in Ref. [40]. Jevicki noticed that the infrared divergences of Feynman diagrams for the ground-state energy of the 2D O($N$) invariant $\sigma$-model, calculated perturbatively up to two loops, cancel exactly when summed over all diagrams. The infrared divergences in 2D models with continuous symmetry stem from the presence of massless Goldstone excitations in the symmetry-broken phase associated with the classical vacuum around which the perturbation is performed. After verifying the cancellation up to second-order perturbation, Elitzur later conjectured that, when summed over all Feynman diagrams, the infrared divergences in any O($N$) invariant field correlations would also cancel to any order perturbation within the low-temperature expansion [41] around the classical vacuum or saddle-point solution. Elitzur

commented that these massless Goldstone excitations are generated by applying symmetry-generating operators to the classical broken vacuum, and since invariant quantities commute with the symmetry generators, they remain unaffected by their application to the vacuum. It is thus expected that invariant quantities will decouple from Goldstone excitations and may consequently remain finite. This conjecture was later proven up to arbitrary perturbative orders by David [42], demonstrating that the cancellation of infrared divergences in $O(N)$ invariant functions occurs at non-exceptional momenta to any perturbative order.

The same idea was subsequently successfully applied to the vortex physics of type-II superconductors around the year 2000. Almost sixty years ago, Eilenberger calculated the spectrum of harmonic excitations of the Abrikosov vortex lattice based on the Ginzburg-Landau theory of type-II superconductors in an external magnetic field [43]. Maki and Takayama later pointed out that the gapless mode is softer than the conventional Goldstone mode expected from the spontaneous breaking of both translational and U(1) symmetries, leading to infrared divergences in the perturbation expansion around the Abrikosov vortex lattice state [44]. The cancellation of infrared divergences in the effective free energy up to two-loop order was first noted in Ref. [45], but the final result was obtained in Refs. [46, 47] after including the Umklapp contribution in two-loop diagrams. Therefore, the infrared divergences most probably cancel for any U(1) invariant function including the effective potential, and even for the Ginzburg-Landau theory of type-II superconductors in an external magnetic field. Although U(1) invariance is restored due to the massless excitation (acoustic excitation mode), and although the infrared divergence cancels for U(1) invariant quantities such as the structure function as shown in Ref. [47], the breaking of translational symmetry (from the homogeneous solution to the vortex lattice solution) remains. The phase transition corresponding to this translational symmetry breaking is typically first-order. For vortex matter, it is referred to as the vortex melting transition. By comparing the effective free energy of the vortex solid obtained in Refs. [46, 47] with that of the vortex liquid, Refs. [48, 49] determined the melting line of the vortex lattice. The effective free energy of liquid was obtained via Borel-Padé resummation of the expansion series (up to nine-loop diagrams) given in Ref. [49]. Details of the theoretical calculations can be found in Refs. [50, 51]. Subsequent experiments have verified the theoretical melting line predictions [52, 53].

In view of the literature review above, both many-body theory and statistical field theory for 2D systems share similar difficulties, and it is beneficial to borrow ideas from each other. For the Hubbard model relevant to cuprate high-$T_c$ superconductors, the most relevant coupling in the cuprates is the strong-coupling $U \geq 8$. The phase transition temperatures (in unit of tunnelling amplitude $t$) are generally very small, typically $T \leq 1/30$, or equivalently, inverse temperatures $\beta \geq 30$. Seeking a many-body theory to tackle the challenges for 2D systems with continuous symmetry, such as the 2D Hubbard model with the spin rotational symmetry, at strong coupling and low temperature is therefore a necessity. Even if we can address the problem using DiagMC calculations for the 2D Hubbard model at very low temperatures, we speculate that $U$ cannot be as large as 8 due to the perturbative nature of DiagMC (the Slater-like branch), and thus it cannot capture the physics of the strong-coupling regime (the Mott-Heisenberg branch).

## 1.4 Motivation for Non-perturbative Many-Body Methods

In this paper, our main task is seeking many-body theories that are free from the limitations of numerical methods, such as the sign problem or huge computational cost. However, since any many-body theory is intrinsically approximate rather than exact, we should be cautious in selecting a many-body approximation method.

Although we cannot maintain all these fundamental relations in a many-body method, we should aim to minimize the violation of these identities as far as possible. It is difficult to

preserve all identities, so we can only maintain one of them. Given the importance of the WTI (since experimental consistency is verified by the $f$-sum rules), it is better to preserve the WTI due to its foundation in conservation laws. Although the $\chi$-sum rule is not our primary requirement, it is desirable to keep its violation small for any reasonable approximation. The simplest fermionic many-body method is the HF approximation, which also known as mean-field theory for fermionic many-body systems. For this approximation, the response function related to both the two-body correlator and the vertex function is given by the RPA formula, and the WTI is satisfied. However, the $\chi$-sum rule is badly violated in the strongly correlated Hubbard model. Therefore, many-body theory should go beyond the HF approximation in order to minimize the violation of the $\chi$-sum rule for strongly correlated systems such as the Hubbard model.

For statistical field theories such as Ginzburg-Landau theory, mean-field theory corresponds to the perturbative expansion around the saddle point of the free energy. However, non-perturbative vortex physics lies beyond such a perturbative mean-field description, necessitating methods that go beyond mean-field theory within the framework of statistical field theory [51]. The simplest non-perturbative approach in statistical field theory is the self-consistent HF theory (equivalent to Gaussian variational theory). The spinodal line was predicted and located within this framework [51], and several experimental studies have confirmed this non-perturbative result [54–56]. However, in the broken phase (where the expectation of order parameter field is nonzero), HF theory violates the Goldstone theorem, which predicts the existence of a massless mode in the spontaneous continuous symmetry breaking phase. The massless mode can be recovered using the covariance theory for calculating correlation functions of the order parameter field, for example, the correlator in the vortex lattice state [57]. Further discussion will be provided in the main text of this paper.

Substantial efforts and progress have also been made in the development of self-consistent many-body approximation theories. Several versions of two-particle self-consistent approaches have been proposed [16–19]. For example, at temperature $T = 0.2$, the results in Refs. [16–19] agree reasonably well with benchmark Monte Carlo data for $U \leq 2.5$. However, the validity of these approaches at larger $U$ and lower temperatures remains unknown. Furthermore, it is unclear whether their results are consistent with the WTI or not.

DQMC can provide exact results in regimes where the sign problem is not serious. However, at low temperatures and strong coupling, a serious sign problem occurs at finite doping, precisely where the high-$T_c$ superconducting phase appears. Consequently, no reliable data are available in these most interesting regions. In contrast, at half-filling, no sign problem occurs, and non-perturbative exact results from DQMC are accessible even at low temperatures and strong coupling [58]. These results should serve as benchmarks for testing many-body theories under development. Only after a many-body theory has passed such benchmark tests against available DQMC data in sign-problem-free regions can we confidently apply it to the regions affected by a severe sign problem.

The covariance theory was developed in Ref. [59], where the response function along with one- and two-body correlators satisfy both FDR and WTI for a generic many-body approximation theory. The HF approximation for a fermionic many-body system is sometimes referred to as a mean-field theory, as fluctuations of the order parameter field are not considered within this approximation. In contrast, the HF approximation for a statistical field theory (or for bosonic systems) goes beyond mean-field theory, as fluctuations of the order parameter field are included, and only the saddle-point solution constitutes the mean-field theory. In the HF-covariance approach (i.e., application of the covariance theory to HF theory or mean-field theory for fermionic many-body systems), two-body correlations or susceptibilities are given by an RPA-like formula. However, we find that the HF results seriously violate the $\chi$-sum rule even at weak coupling $U = 2$ and temperature $T = 0.2$ at half-filling for the Hubbard model.

Nevertheless, reasonable results can still be obtained within HF when the temperature is far away from the mean-field transition temperature, and valuable lessons can be drawn from Ref. [60].

Using the HF approximation and the post-Hartree-Fock (PHF) approximation, which incorporates self-energy corrections up to two-loop, calculations of the Green's function were performed for the half-filled Hubbard model in 2D and compared with DQMC simulations. The imaginary part of the Green's function $G(\tau, \boldsymbol{k})$ is evaluated at $\boldsymbol{k} = (\pi, 0)$ on a $12 \times 12$ lattice. The temperature is fixed at $T = 1$, with coupling strengths $U = 1, 4, 6, 8, 12$. A spurious mean-field transition occurs around $U_c = 4.9$. It is found that at weak coupling ($U = 1, 4$) below $U_c$, the agreement with DQMC is excellent when the Green's function is perturbatively corrected within PGA. Above $U_c$, symmetrized Green's functions from symmetry-broken solutions obtained within GA and PGA are presented and compared with DQMC. For intermediate coupling $U = 6$ just above $U_c$, both GA and PGA show significant deviations from DQMC. However, for stronger couplings $U = 8, 12$ (compared to $U_c$), the agreement improves sufficiently far from the critical coupling. Thus, both the HF and PHF approximation perform poorly in a large regime around the mean-field critical temperature or critical coupling. Reasonable results are given only when far from the mean-field critical regime.

Based on the identified issues and the experience and lessons drawn, we will implement the following strategies to tackle the problem. We have to go beyond mean-field theory, which is intrinsically perturbative for strongly correlated systems. One of the simplest non-perturbative approaches beyond the HF (or mean-field) approximation for studying the 2D Hubbard model is the GW theory. We will use the GW theory along with the covariance theory, which can satisfy FDR and WTI. We will assess the violation level of the local moment sum rules (or the $\chi$-sum rule), and compare the results with benchmark data from DQMC simulations of the half-filled 2D Hubbard model on a finite lattice at low temperature and strong coupling. Only after the theory has passed the $\chi$-sum rule test (exhibiting low-level violation) and shows only small deviations from the benchmark results can we have confidence in its validity.

We will show that in the high-temperature phase predicted by the GW-covariance theory, the violation of the $\chi$-sum rule remains small even for quite strong coupling. For the half-filled 2D Hubbard model at strong coupling and low temperature, we will calculate symmetry-invariant one- and two-body correlators in the pseudo AF phase from the GW-covariance approximation along with the covariance theory. Although the covariance theory satisfies the FDR and WTI, the symmetry-invariant correlators of the AF GW state at low temperature will violate the $\chi$-sum rule relation, generally by less than 10% for $U = 4$, except in the overlap region between the high-temperature and low-temperature branches. Even within the overlap regime, a state can be identified that exhibits only a small violation of the $\chi$-sum rule and has short-range behavior consistent with benchmark data. We find that the main deviation arises because the spin susceptibility is significantly large at certain exceptional momentum, where it deviates from the DQMC results. However, for non-exceptional momenta, the spin susceptibility shows very good agreement with that obtained from DQMC. A similar result was noted by David [42] for 2D $\sigma$-models at low temperature, where the invariant correlator at exceptional momentum can diverge. In a finite lattice system, such an invariant correlator at exceptional momentum is on the order of some power of the number of lattice sites, which is huge but not divergent. In practice, we use the $\chi$-sum rule to improve the accuracy of the correlators at the exceptional momentum point. Upon applying the $\chi$-sum rule correction, the short-range behavior of the spin correlation exhibits remarkable agreement with the DQMC benchmark results.

## 1.5 Organization of this article

This article is organized as follows: In Sec. 2, we establish the symmetrization scheme. In Sec. 3.1 and Sec. 3.2, we introduce the covariance theory and GW approximation. In Sec. 3.3, Sec. 3.4 and Sec. 3.5, we implement the symmetrization scheme and the GW-covariance approximation to the 2D Hubbard model on the A-B lattice; this approach predicts a pseudo paramagnetic-AF phase transition in the 2D Hubbard model, breaking the spin SU(2) symmetry. In Sec. 4, we present the benchmark of our approach with the DQMC results. In Sec. 5, we suggest a self-consistency criterion for many-body approximation methods. Finally, in Sec. 6, we provide a brief summary and discussion.

## 2 General symmetrization scheme

In the context of many-body physics, physical quantities can be expressed as statistical ensemble averages of certain physical functionals of the fields. For example, such a functional can represent density-density fields, whose statistical ensemble average is referred to as the density-density correlation. The group action/transformation is defined by specifying how the fundamental fields, such as the electron field, transform under each group element. If the original action remains invariant under transformations of the fields by a group $\mathcal{G}$, the system is said to be $\mathcal{G}$-invariant or $\mathcal{G}$-symmetric. At high temperatures, such a system is typically in what is known as a $\mathcal{G}$-invariant state, in which all physical quantities (or physical functionals of the fields) are invariant under $\mathcal{G}$. However, at low temperatures, the state that minimizes the free energy may not be $\mathcal{G}$-invariant, in which case certain physical functionals of the fields are not invariant under $\mathcal{G}$. When this happens, the system is said to be in a spontaneous symmetry-breaking phase. For example, the Hubbard model is SU(2)-invariant as its action remains unchanged under SU(2) transformations acting on the spin indices of the electron fields. At high temperatures, the half-filled 3D Hubbard model with interaction is in a SU(2)-invariant paramagnetic state, while at low temperatures it is in an AF-symmetry-breaking phase. In the latter, certain physical functionals of the fields, such as the AF order parameter field, break the SU(2) symmetry.

In principle, symmetry breaking in low-dimensional systems may be summarized as follows. For any finite lattice, there is neither spontaneous symmetry breaking nor phase transition, but only a crossover between the low- and high-temperature regimes. Even in infinite lattice systems, continuous symmetry breaking does not occur in 2D. For continuous systems in the O($N$) universality class, there is similarly neither continuous symmetry breaking nor phase transition, but only a crossover between the low- and high-temperature regimes. The only exception is the O(2) case, for which a finite-temperature BKT phase transition can occur in either continuous or infinite lattice systems.

In practice, however, many-body approximations may contain continuous symmetry-breaking solution in 2D finite systems. This contradiction prevents the application of many-body theory at temperatures below the instability point of the group-invariant state. To tackle this difficulty, we introduce the symmetrization scheme. As noted by Jevicki et al., even though continuous symmetry breaking does not occur in 2D at low temperatures, one should still begin from a symmetry-broken state, and the symmetry-invariant correlators and other symmetry-invariant physical quantities calculated within this symmetry-broken framework can nevertheless give reasonable quantitative estimates. The idea of Jevicki et al. has been generalized in a way somewhat different from the original one to study the Hubbard model at half-filling and low temperature, and this generalized framework was confirmed to be valid by comparison with DQMC results [60]. Specifically, we can calculate not only symmetry-invariant physical quantities in the pseudo symmetry-breaking state but also those non-invariant quantities through

the symmetrization. In the following, we outline a general framework for performing symmetrization.

We consider an arbitrary field functional, which may involve one- or two-body fields. For simplicity in conveying our ideas, we consider a finite lattice with a discrete symmetry group $\mathcal{G}$. Physical quantities can be approximately expressed as $F \equiv \langle \Psi | \mathcal{F}(\hat{\psi}_\alpha, \hat{\psi}^\dagger_\beta) | \Psi \rangle$, where $\hat{\psi}_\alpha$ and $\hat{\psi}^\dagger_\beta$ are electron field operators, $\Psi$ is a $\mathcal{G}$-symmetry-breaking state with the lowest free energy among all possible configurations, and $F$ represents the statistical ensemble average of the field-operator functional $\mathcal{F}$ over the state $\Psi$. Due to symmetry, for any $g \in \mathcal{G}$, the state $g\Psi$ also has the lowest energy, just as $\Psi$ does. However, $g\Psi$ and $\Psi$ are distinct states, corresponding to different order parameters. Since the lattice is finite, there is no infinite energy barrier between these states. Consequently, tunneling between them is possible. Thus, all states in $\{g\Psi \,|\, g \in \mathcal{G}\}$ may contribute to physical quantities, and the physical quantities should be expressed by the average value of the field functional over all possible symmetry-breaking states. More precisely, we should consider

$$\bar{F} \equiv \frac{1}{|\mathcal{G}|} \sum_{g \in \mathcal{G}} \langle g\Psi | \mathcal{F}(\hat{\psi}_\alpha, \hat{\psi}^\dagger_\beta) | g\Psi \rangle, \tag{1}$$

where $|\mathcal{G}|$ denotes the number of group $\mathcal{G}$ elements. This is equivalent to $\bar{F} = \langle \Psi | \bar{\mathcal{F}}(\hat{\psi}_\alpha, \hat{\psi}^\dagger_\beta) | \Psi \rangle$ with

$$\bar{\mathcal{F}}(\hat{\psi}_\alpha, \hat{\psi}^\dagger_\beta) \equiv \frac{1}{|\mathcal{G}|} \sum_{g \in \mathcal{G}} \mathcal{F}(g^\dagger \hat{\psi}_\alpha g, g \hat{\psi}^\dagger_\beta g^\dagger), \tag{2}$$

where $g^\dagger \hat{\psi}_\alpha g$ and $g \hat{\psi}^\dagger_\beta g^\dagger$ represent the field transformations under the group $\mathcal{G}$. The formula states that, for certain group, the physical quantity corresponds to the statistical ensemble average of the group-symmetric field functional over one group-symmetry-breaking state. Since the $\bar{\mathcal{F}}(\hat{\psi}_\alpha, \hat{\psi}^\dagger_\beta)$ is $\mathcal{G}$-symmetric, its statistical ensemble average $\bar{F}$ over any spontaneous $\mathcal{G}$-symmetry-breaking state is the same. In this case, the order parameter field will vanish after this symmetrization, while other quantities, such as the spin correlation function (as will be shown later), will become independent of the symmetry-breaking direction.

When dealing with continuous symmetries, the basic idea of taking average remains unchanged only the summation is replaced by an integral, i.e., the invariant Haar measure [61]. Specifically, if $U$ represents the matrix representation of an element from a continuous symmetry group $\mathcal{G}$, the symmetrization procedure involves integration over the group manifold with the appropriate Haar measure

$$\overline{\left\langle \hat{\psi}^\dagger_{\alpha_1} \dots \hat{\psi}^\dagger_{\alpha_n} \hat{\psi}_{\beta_1} \dots \hat{\psi}_{\beta_m} \right\rangle} = \int \mathrm{d}U \, [U^*]^{\alpha'_1}_{\alpha_1} \dots [U^*]^{\alpha'_n}_{\alpha_n} U^{\beta'_1}_{\beta_1} \dots U^{\beta'_m}_{\beta_m} \left\langle \hat{\psi}^\dagger_{\alpha'_1} \dots \hat{\psi}^\dagger_{\alpha'_n} \hat{\psi}_{\beta'_1} \dots \hat{\psi}_{\beta'_m} \right\rangle, \tag{3}$$

which is a generalization of Eq. (1) for continuous symmetric groups.

In summary, for any finite lattice with either discrete or continuous symmetry, as well as any 2D infinite lattice with continuous symmetry, symmetry-breaking solutions are not problematic, provided that the field functional is symmetrized first. Physically speaking, we envisage that these systems exist in the form of different **domains** inside the sample, each characterized by their own "order parameters". Inside a domain, fluctuations are considerable, yet the "order parameter" remains identifiable. On the scale of the entire sample, these "order parameters" behave as short-range fluctuations. Such "order parameters" can indeed be approximately observed experimentally [62]. However, long-range correlations between different domains are very weak, which corresponds to the absence of long-range order [39]. The symmetrization scheme is precisely the averaging over all domains from a macroscopic perspective. Formally speaking, the symmetrization scheme is taking the average of the field functional $\mathcal{F}$ over all the

group action $g$ acting on the electron fields. This is equivalent to averaging the field functional $\mathcal{F}$ over all symmetry-breaking states. The thermodynamic quantities of the system, particularly the free energy, remain invariant across different spontaneous symmetry-breaking states, and thus these states should be assigned equal weight to justify our symmetrization scheme.

In this work, we investigate the paramagnetic-to-AF phase transition (or crossover) using many-body approximations in the Hubbard model on the 2D square lattice, as an example to specifically illustrate the application of the symmetrization scheme (see Sec. 3.5). This pseudo phase transition involves the breaking of both discrete translational symmetry and continuous SU(2) spin symmetry, and the continuous SU(2) spin symmetry needs to be restored via symmetrization.

# 3 Formalism

In this work, we will present the theories within the framework of functional integrals [63–65]. We consider a generalized system with the Matsubara action:

$$
\mathcal{S}[\psi^*, \psi] = -\sum_{\alpha_1 \alpha_2} \int d(12)\, \psi^*_{\alpha_1}(1) T_{\alpha_1 \alpha_2}(1,2) \psi_{\alpha_2}(2)
$$
$$
-\frac{1}{2} \sum_{ab} \int d(12) S^a(1) V^{ab}(1,2) S^b(2),
\tag{4}
$$

where $\alpha = \uparrow, \downarrow$ means spin up and down, $\sigma^0$ is the identity matrix, $\sigma^a$ ($a = 0, x, y, z$) are Pauli matrices, and the

$$
S^a(1) = \sum_{\alpha\beta} \psi^*_\alpha(1) \sigma^a_{\alpha\beta} \psi_\beta(1)
\tag{5}
$$

represents charge operator for $a = 0$ and spin operator for $a = x, y, z$, relatively. $\psi^*, \psi$ are Grassmannian fields. The numbers in parentheses denote different spacetime coordinates, as $(1) \doteq (\tau_1, \boldsymbol{x}_1)$, $\int d(1) \doteq \int_0^\beta d\tau_1 \sum_{\boldsymbol{x}_1}$, where $\beta$ is the inverse temperature, $0 \leq \tau_1 < \beta$ is the Matsubara time, $\boldsymbol{x}_1$ is the space coordinate. $T_{\alpha_1 \alpha_2}(1,2)$ is the quadratic term of the action,

$$
T_{\alpha_1 \alpha_2}(1,2) = \delta_{\alpha_1 \alpha_2} \delta(\tau_1, \tau_2) \delta_{\boldsymbol{x}_1, \boldsymbol{x}_2}(-\partial_{\tau_2}) - \mathcal{K}_{0\alpha_1 \alpha_2}(1,2)
\tag{6}
$$

where $\mathcal{K}_{0\alpha_1 \alpha_2}(1,2)$ corresponds to the kinetic term of the grand Hamiltonian in Matsubara representation. $V^{ab}(1,2)$ is the interaction satisfying $V^{ab}(1,2) = V^{ba}(2,1)$. The grand partition function is

$$
\mathcal{Z} = \int \mathcal{D}[\psi^*, \psi]\, e^{-\mathcal{S}[\psi^*, \psi]}.
\tag{7}
$$

Definition of the one-body Green's function is

$$
G_{\alpha_1 \alpha_2}(1,2) = -\langle \psi_{\alpha_1}(1) \psi^*_{\alpha_2}(2) \rangle,
\tag{8}
$$

where $\langle \ldots \rangle = \mathcal{Z}^{-1} \int \mathcal{D}[\psi^*, \psi] \ldots e^{-\mathcal{S}}$ is the ensemble average. For convenience, denote the spin structure by matrix formation:

$$
\boldsymbol{G} \doteq \begin{pmatrix} G_{\uparrow\uparrow} & G_{\uparrow\downarrow} \\ G_{\downarrow\uparrow} & G_{\downarrow\downarrow} \end{pmatrix}.
\tag{9}
$$

The matrix product is $[\boldsymbol{X}\boldsymbol{Y}]_{\alpha_1 \alpha_3} = \sum_{\alpha_2} X_{\alpha_1 \alpha_2} Y_{\alpha_2 \alpha_3}$, and the trace is $\text{Tr}[\boldsymbol{G}] = G_{\uparrow\uparrow} + G_{\downarrow\downarrow}$. The inverse of the Green's function is defined by

$$
\sum_{\alpha_2} \int d(2)\, G^{-1}_{\alpha_1 \alpha_2}(1,2) G_{\alpha_2 \alpha_3}(2,3) = \delta_{\alpha_1 \alpha_3} \delta(1,3),
\tag{10}
$$

thus the non-interacting Green's function is $G_{0\alpha_1\alpha_2}(1,2) = T^{-1}_{\alpha_1\alpha_2}(1,2)$. Definition of the connected two-body correlation function is

$$\chi_{XY}(1,2) = \langle X(1)Y(2)\rangle_C, \tag{11}$$

where $X, Y$ are one-body operators, and $\langle\ldots\rangle_C$ represents the connected-diagram contribution in the ensemble average.

The following subsections are organized as follows: Present the covariance theory as a general framework for treating two-body correlation functions. Describe the GW approximation for calculating single-particle Green's functions, along with the GW-covariance approximation for obtaining two-body correlation functions. Specialize the generalized action in Eq. (4) to the case of the 2D Hubbard model and introduce the A-B lattice to describe the AF state. Apply the symmetrization scheme to this particular case.

## 3.1 Covariance theory

A reliable computation of two-body correlation functions is crucial for testing the symmetrization scheme. Specifically, the approximate correlation function should adhere as closely as possible to fundamental relations such as the FDR and WTI, and the Pauli exclusion principle. To address this issue, we introduce the covariance theory [59]. The key of the covariance theory is to systematically construct the covariance equations (refer to Eq. (16)) based on an approximation method for one-body Green's function. By solving the covariance equations to obtain the vertex $\Lambda$, this framework yields two-body correlation functions (see Eq.(13)). The resulting two-body correlation functions will inherently satisfy the FDR, as the latter is derived by Kubo via functional differentiation in the context of response theory [11–14] and thus coincides with Eq. (13). No specific requirements are imposed on the form of the self-energy in Eq. (16), thus one can apply this framework on various approximation methods, as long as the resulting covariance equations are solvable. As an application example, Li et al. present the GW-covariance approximation, in which the resulting correlation function is demonstrated to satisfy the WTI [59]. Although this proof is limited to the GW-covariance approximation, it is reasonable to conjecture that the correlation functions yielded by the covariance theory will always respect the WTI, provided the underlying one-body approach possesses the corresponding symmetry.

We start with the definition of connected correlation function Eq. (11). In general, an one-body operators $X$ have such quadratic structure as

$$X(3) = \sum_{\alpha_1\alpha_2}\int d(12)\psi^*_{\alpha_1}(1)K_{X\alpha_1\alpha_2}(1,2;3)\psi_{\alpha_2}(2), \tag{12}$$

and we term $K_X$ the kernel of $X$. The spin operator $S^a(3)$, for instance, corresponds to $K_{S^a}(1,2;3) = \sigma^a\delta(1,2)\delta(1,3)$. We consider an external source $\phi$ which is coupled to the operator $Y$, that is, modify the action $\mathcal{S}$ to $\mathcal{S} - \int d(3)\phi(3)Y(3)$. The correlation function is then given by

$$\chi_{XY}(1,2) = \frac{\delta\langle X(1)\rangle}{\delta\phi(2)} = \int d(34)\mathrm{Tr}\big[K_X(3,4;1)\Lambda_\phi(4,3;2)\big]. \tag{13}$$

Here, we denote $\delta G(1,2)/\delta\phi(3)$ by $\Lambda_\phi(1,2;3)$, and denote $\delta G^{-1}(1,2)/\delta\phi(3)$ by $\Gamma_\phi(1,2;3)$. They can be related to each other by

$$\Lambda_\phi(1,2;3) = -\int d(45)G(1,4)\Gamma_\phi(4,5;3)G(5,2). \tag{14}$$

The problem now becomes how to calculate the vertex $\Lambda_\phi$ (or $\Gamma_\phi$). Notice that the external source is of the same kind with the kinetic term in action, thus modifying the action as $S \to S - \int \mathrm{d}(3)\phi(3)Y(3)$ is equivalent to modifying the non-interacting Green's function as $G_0 \to G_0[\phi]$, i.e.,

$$G_0^{-1}[\phi](1,2) = G_0^{-1}(1,2) + \int \mathrm{d}(3)\phi(3)K_Y(1,2;3). \tag{15}$$

Therefore, in the presence of the external source $\phi$, as long as the modification $G_0 \to G_0[\phi]$ is made, the Dyson equation $G^{-1}(1,2) = G_0^{-1}[\phi](1,2) - \Sigma(1,2)$ remains valid. This allows one to systematically construct the covariance equations for any given perturbative or self-consistent theory, as

$$\Gamma_\phi(1,2;3) = \gamma_\phi(1,2;3) - \frac{\delta\Sigma(1,2)}{\delta\phi(3)}, \tag{16}$$

where

$$\gamma_\phi(1,2;3) \equiv \frac{\delta G_0^{-1}[\phi](1,2)}{\delta\phi(3)} = K_Y(1,2;3). \tag{17}$$

The covariance equations are governed solely by the kernel $K_Y$, and their computational cost is independent of the kernel choice. Consequently, solving these equations is generally tractable, provided the approximate self-energy used in Eq. (16) is not excessively complicated.

## 3.2 The GW and GW-covariance approximation

The GW approximation, a non-perturbative method for calculating one-body Green's function, was proposed by Hedin in 1965 [66]. To construct a spin-dependent GW approximation, one employs a generalized formalism [67].

For the action in Eq. (4), one can then derive a set of equations called Hedin's equations. In these equations, the self-energy $\Sigma$ comprises two terms, $\Sigma(1,2) = \Sigma_H(1,2) + \Sigma'(1,2)$. The first is the Hartree self energy,

$$\Sigma_H(1,2) = -\delta(1,2)\sum_{ab}\int \mathrm{d}(3)\sigma^a V^{ab}(1,3)\mathrm{Tr}[\sigma^b G(3,3)]. \tag{18}$$

The second is

$$\Sigma'(1,2) = \sum_{ab}\int \mathrm{d}(34)\sigma^a G(1,3)\Gamma_H^b(3,2;4)W^{ba}(4,1), \tag{19}$$

with

$$[W^{-1}]^{ab}(1,2) = [V^{-1}]^{ab}(1,2) - P^{ab}(1,2), \tag{20}$$

$$P^{ab}(1,2) = -\int \mathrm{d}(34)\mathrm{Tr}\big[\sigma^a G(1,3)\Gamma_H^b(3,4;2)G(4,1)\big], \tag{21}$$

where $\Gamma_H$ is the Hedin's vertex function. The Dyson equation $G^{-1}(1,2) = G_0^{-1}(1,2) - \Sigma(1,2)$ and Eqs.(18, 19, 20, 21) are precisely the Hedin's equations. The formal definition of $\Gamma_H$ and details derivations are provided in Appendix A. By introducing approximations order by order, the Hedin's equations serve as a bridge connecting the exact many-body theory and practical calculations. Among these approximations, the GW approximation retains the leading order term of the vertex function $\Gamma_H^a(1,2;3) \simeq \sigma^a\delta(1,2)\delta(1,3)$. Under GW approximation, Eq.(19, 21) becomes

$$\Sigma_{GW}(1,2) = \sum_{ab}\sigma^a G(1,2)\sigma^b W^{ba}(2,1), \tag{22}$$

$$P^{ab}(1,2) = -\mathrm{Tr}\big[\sigma^a G(1,2)\sigma^b G(2,1)\big], \tag{23}$$

relatively. The combination of Dyson's equation and Eqs.(18, 22, 20, 23) comprise the GW equations, and can be self-consistently solved to obtain the Green's function.

Applying the covariance theory on the GW approximation (substituting the GW approximate self-energy $\Sigma = \Sigma_H + \Sigma_{GW}$ into Eq. (16)), one obtains

$$\Gamma_\phi = \gamma_\phi - \Gamma_\phi^H - \Gamma_\phi^{MT} - \Gamma_\phi^{AL}. \tag{24}$$

The $\Gamma_\phi^H$ is the functional derivative of the Hartree self energy,

$$
\begin{aligned}
\mathbf{\Gamma}_\phi^H(1,2;3) &= \frac{\delta \mathbf{\Sigma}_H(1,2)}{\delta \phi(3)} \\
&= -\delta(1,2) \sum_{ab} \int \mathrm{d}(3) \boldsymbol{\sigma}^a V^{ab}(1,4) \mathrm{Tr}[\boldsymbol{\sigma}^b \mathbf{\Lambda}(4,4;3)].
\end{aligned} \tag{25}
$$

The $\Gamma_\phi^{MT}$ and $\Gamma_\phi^{AL}$ come from $\Sigma_{GW}$,

$$\mathbf{\Gamma}_\phi^{MT}(1,2;3) + \mathbf{\Gamma}_\phi^{AL}(1,2;3) = \frac{\delta \mathbf{\Sigma}_{GW}(1,2)}{\delta \phi(3)}. \tag{26}$$

They are respectively

$$\mathbf{\Gamma}_\phi^{MT}(1,2;3) = \sum_{ab} \boldsymbol{\sigma}^a \mathbf{\Lambda}_\phi(1,2;3) \boldsymbol{\sigma}^b W^{ba}(2,1), \tag{27}$$

$$\mathbf{\Gamma}_\phi^{AL}(1,2;3) = -\sum_{abcd} \boldsymbol{\sigma}^a \boldsymbol{G}(1,2) \boldsymbol{\sigma}^b \int \mathrm{d}(45) W^{bc}(2,4) \Gamma_\phi^{Wcd}(4,5;3) W^{da}(5,1), \tag{28}$$

$$\Gamma_\phi^{Wcd}(4,5;3) = \mathrm{Tr}\left[\boldsymbol{\sigma}^c \mathbf{\Lambda}_\phi(4,5;3) \boldsymbol{\sigma}^d \boldsymbol{G}(5,4) + \boldsymbol{\sigma}^c \boldsymbol{G}(4,5) \boldsymbol{\sigma}^d \mathbf{\Lambda}_\phi(5,4;3)\right]. \tag{29}$$

Eqs. (24, 17, 25, 27 ,28, 29, 14) are the GW-covariance equations. Knowing the $G$ and $W$, they can be solved self-consistently like solving the GW equations to obtain $\Lambda_\phi$. The two-body correlation function is then given via Eq. (13).

## 3.3 Model

We apply the theory on the 2D repulsive Hubbard model. The grand Hamiltonian of the Hubbard model is

$$\hat{\mathcal{K}} = -\sum_{\langle i,j \rangle \alpha} \left(t_{ij}\hat{c}_{i\alpha}^\dagger \hat{c}_{j\alpha} + h.c.\right) - \mu \sum_{i\alpha} \hat{n}_{i\alpha} + U \sum_i \hat{n}_{i\uparrow} \hat{n}_{i\downarrow}, \tag{30}$$

where $i$ denotes the lattice site, $\alpha = \uparrow, \downarrow$ denotes the spin, $\hat{c}_{i\alpha}^\dagger$ ($\hat{c}_{i\alpha}$) creates (annihilates) a fermion with spin $\alpha$ on site $i$, $\hat{n}_{i\alpha} \equiv \hat{c}_{i\alpha}^\dagger \hat{c}_{i\alpha}$ denotes spin-resolved density operator, $t_{ij}$ denotes the hopping amplitude from site $j$ to site $i$ (with the property $t_{ij} = t_{ji}$), $\mu$ is the chemical potential, and $U > 0$ denotes strength of the on-site repulsive interaction. The doping level of the system can be altered by adjusting the chemical potential, with $\mu$ being precisely $U/2$ in the case of a half-filled system.

Using the Fierz relation

$$\hat{n}_{i\uparrow} \hat{n}_{i\downarrow} = \frac{1}{2} \sum_\alpha \hat{n}_{i\alpha} - \frac{1}{6} \sum_{a=x,y,z} \hat{S}_i^a \hat{S}_i^a, \tag{31}$$

the Hubbard Hamiltonian becomes

$$\hat{\mathcal{K}} = -\sum_{\langle i,j \rangle \alpha} \left(t_{ij}\hat{c}_{i\alpha}^\dagger \hat{c}_{j\alpha} + h.c.\right) + \left(\frac{U}{2} - \mu\right) \sum_{i\alpha} \hat{n}_{i\alpha} - \frac{U}{6} \sum_i \sum_{a=x,y,z} \hat{S}_i^a \hat{S}_i^a. \tag{32}$$

Preserving the spin SU(2) symmetry when doing approximation, we second-quantize the grand Hamiltonian Eq. (32), which is equivalent to substituting

$$\mathcal{K}_0(1,2) = \boldsymbol{\sigma}^0 \delta(\tau_1,\tau_2)\big(-t_{\boldsymbol{x}_1,\boldsymbol{x}_2} - \mu\,\delta_{\boldsymbol{x}_1,\boldsymbol{x}_2}\big), \tag{33}$$

$$V^{ab}(1,2) = \frac{U}{3}\delta(1,2)\sum_{c=x,y,z}\delta^{ac}\delta^{bc} \tag{34}$$

into the quadratic term Eq. (6) of the Matsubara action Eq. (4).

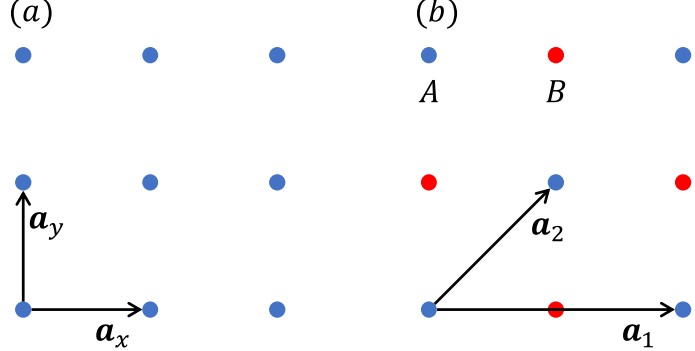

Figure 1: Real-space lattice configurations for paramagnetic and antiferromagnetic solutions of the 2D Hubbard model. (a) Lattice configuration of the paramagnetic solution, with lattice vectors $\boldsymbol{a}_x = (a,0)$ and $\boldsymbol{a}_y = (0,a)$. All sites are equivalent. (b) Lattice structure with AB sublattices for the antiferromagnetic solution with Néel order (A sublattice: blue; B sublattice: red), spanned by $\boldsymbol{a}_1 = (2a,0)$ and $\boldsymbol{a}_2 = (a,a)$.

In this work, we restrict the system to the 2D square lattice. As shown in Fig. 1 (a), the lattice sites located on $\boldsymbol{x} = n_x\boldsymbol{a}_x + n_y\boldsymbol{a}_y$, with periodic boundary condition, $n_x, n_y \in \{0,\ldots,N_x-1\}$ are integers, $a \equiv |\boldsymbol{a}_x| = |\boldsymbol{a}_y|$ is the distance between the two nearest lattice sites. The reciprocal lattice is given by $\boldsymbol{k} = k_x\boldsymbol{b}_x + k_y\boldsymbol{b}_y$, where $\boldsymbol{a}_i \cdot \boldsymbol{b}_j = \delta_{ij}$ (here $i,j = x,y$), and $k_x, k_y \in \{2\pi i/N_x | i = 0,\ldots,N_x-1\}$.

## 3.4   Application in the AF broken phase

We are interested in the case of intermediate-to-strong $U$ ($2 < U < 8$) [68] and small doping, wherein the AF order represents the most prominent symmetry-breaking phenomenon. However, AF order breaks the spatial translation invariance (precisely, the translation invariance on the 2D square lattice), thus one cannot apply the Fourier transformation directly on the space coordinate. To deal with this problem, we introduce the A-B sublattice, which is a convenience tool for describing the structure of AF order.

As shown in Fig. 1 (b), the lattice unit vectors of the A-B sublattice are $\boldsymbol{a}_1 = 2\boldsymbol{a}_x$ and $\boldsymbol{a}_2 = \boldsymbol{a}_x + \boldsymbol{a}_y$. Each unit cell contains two lattice sites denoted by $l = A, B$, whose relative coordinates are $\boldsymbol{u}_A = \boldsymbol{0}$ and $\boldsymbol{u}_B = \boldsymbol{a}_x = \boldsymbol{a}_1/2$. To construct a lattice whose shape is consistent with the original 2D square lattice, we restrict the boundary condition of the A-B sublattice as $n_1 \in \{0,\ldots,N_1-1\}$ and $n_2 \in \{0,\ldots,N_2-1\}$, where $N_2 = 2N_1 = N_x$. Coordinates of the lattice unit cells are $\boldsymbol{R} = (n_1,n_2) = n_1\boldsymbol{a}_1 + n_2\boldsymbol{a}_2$, where $n_1, n_2$ are integers. The $\boldsymbol{R}$'s have the translation invariance on the A-B sublattice no matter whether the AF order exists.

Given any $\boldsymbol{x}$, there is one and only one set of $n_1, n_2 \in \mathbb{Z}$ and $l = A, B$ such that $\boldsymbol{x} = (n_1,n_2)+\boldsymbol{u}_l$ (with $\boldsymbol{u}_A = \boldsymbol{0}$ and $\boldsymbol{u}_B = \boldsymbol{a}_1/2$). Thus, it's valid to replace the space coordinates $\boldsymbol{x}$ by these new quantum numbers $\boldsymbol{R}, l$. Namely, using the new notations $l_1$ and $1 = (\tau_1,\boldsymbol{R}_1)$ to replace the old one $1 = (\tau_1,\boldsymbol{x}_1)$, and using $\sum_{l_1}\int \mathrm{d}(1) \dot{=} \sum_{l_1}\int_0^\beta \mathrm{d}\tau_1 \sum_{\boldsymbol{R}_1}$ to replace $\int \mathrm{d}(1) \dot{=} \int_0^\beta \mathrm{d}\tau_1 \sum_{\boldsymbol{x}_1}$. For

example, the polarization function Eq. (21) becomes

$$P^{al_1, bl_2}(1, 2) = \text{Tr}\left[\sigma^a G^{l_1, l_2}(1, 2)\sigma^b G^{l_2, l_1}(2, 1)\right], \tag{35}$$

where the Green's function $G(1, 2)$ with $1 = (\tau_1, \boldsymbol{x}_1) = (\tau_1, \boldsymbol{R}_1 + \boldsymbol{u}_{l_1})$ becomes $G^{l_1 l_2}(1, 2)$ with $1 = (\tau_1, \boldsymbol{R}_1)$, $l_1, l_2 = A, B$. Using such quantum numbers, theories can be formulated to be able to describe the AF broken phase.

The momentum space is defined on the reciprocal lattice:

$$\boldsymbol{k} = (k_1, k_2) = k_1 \boldsymbol{b}_1 + k_2 \boldsymbol{b}_2, \tag{36}$$

where $\boldsymbol{b}_1 = (\boldsymbol{b}_x - \boldsymbol{b}_y)/2$, $\boldsymbol{b}_2 = \boldsymbol{b}_y$ are reciprocal unit vectors of the A-B sublattice, with $k_1 \in \{2\pi i/N_1 | i = 0, \ldots, N_1 - 1\}$ and $k_2 \in \{2\pi i/N_2 | i = 0, \ldots, N_2 - 1\}$. Define the Fourier transformation in A-B sublattice as

$$\tilde{F}^{l_1 l_2}(\omega_n, \boldsymbol{k}) = \int_0^\beta \mathrm{d}\tau \sum_{\boldsymbol{R}} \mathrm{e}^{\mathrm{i}\omega_n \tau - \mathrm{i}\boldsymbol{k}\cdot\boldsymbol{R}} F^{l_1 l_2}(\tau, \boldsymbol{R}), \tag{37}$$

$$F^{l_1 l_2}(\tau, \boldsymbol{R}) = \frac{1}{\beta N} \sum_{\omega_n, \boldsymbol{k}} \mathrm{e}^{-\mathrm{i}\omega_n \tau + \mathrm{i}\boldsymbol{k}\cdot\boldsymbol{R}} \tilde{F}^{l_1 l_2}(\omega_n, \boldsymbol{k}), \tag{38}$$

where $\beta$ is the inverse temperature, $N$ is the number of unit cells of the A-B sublattice, and the translation invariance of the A-B sublattice, $F^{l_1 l_2}(\tau_1 - \tau_2, \boldsymbol{R}_1 - \boldsymbol{R}_2) = F^{l_1 l_2}(1, 2)$, has been implied. The GW and GW-covariance equations in momentum space are given in Appendix B. The Hubbard model can be implemented by the following kinetic term and interaction term,

$$\tilde{K}_0^{l_1 l_2}(\omega_n, \boldsymbol{k}) = \sigma^0\left(-\tilde{t}_{\boldsymbol{k}}^{l_1 l_2} - \mu\,\delta_{l_1 l_2}\right), \tag{39}$$

$$\tilde{V}^{al_1, bl_2}(\omega_n, \boldsymbol{k}) = \frac{U}{3}\delta_{l_1 l_2} \sum_{c=x, y, z} \delta^{ac}\delta^{bc}. \tag{40}$$

We only consider the nearest hoping $t_{\boldsymbol{x}, \boldsymbol{x}\pm\boldsymbol{a}_x} = t_{\boldsymbol{x}, \boldsymbol{x}\pm\boldsymbol{a}_x} = t$. In momentum space, such hopping amplitude is

$$\tilde{t}_{\boldsymbol{k}}^{AA} = \tilde{t}_{\boldsymbol{k}}^{BB} = 0, \tag{41}$$

$$\tilde{t}_{\boldsymbol{k}}^{AB} = \tilde{t}_{\boldsymbol{k}}^{BA*} = 2t\left[\cos\left(\frac{k_1}{2}\right) + \cos\left(k_2 - \frac{k_1}{2}\right)\right]\mathrm{e}^{-\mathrm{i}k_1/2}. \tag{42}$$

### 3.5  Symmetrization for SU(2) broken AF phase

The spatial distribution of the AF order parameter can be expressed as follows. Without loss of generality, it is assumed that the direction of the magnetic moment $\langle \boldsymbol{S}(\boldsymbol{x}) \rangle$ is along the $z$-axis.

$$\langle S^x(\boldsymbol{x}) \rangle = \langle S^y(\boldsymbol{x}) \rangle = 0, \quad \langle S^z(\boldsymbol{x}) \rangle = S_0^z\,\mathrm{e}^{\mathrm{i}\boldsymbol{Q}\cdot\boldsymbol{x}} = \sum_{\boldsymbol{k}\in\mathcal{B}} S_0^z \delta_{\boldsymbol{k}, \boldsymbol{Q}}\mathrm{e}^{\mathrm{i}\boldsymbol{k}\cdot\boldsymbol{x}}, \tag{43}$$

where $S_0^z$ is amplitude of the AF wave, $\boldsymbol{Q} = \pi\boldsymbol{b}_x + \pi\boldsymbol{b}_y$ is the AF wave vector, and we denote the reciprocal lattice of 2D square lattice as $\mathcal{B}$ for clarity. The amplitude of AF order breaks SU(2) invariance and the wave vector $\boldsymbol{Q}$ is related to the breaking of phase translation invariance. We only need to restore the continuous symmetry, SU(2) symmetry. However, specially for the AF order, the effect of the wave vector ceases to exist when the amplitude disappears, the translation invariance caused by the AF order happens to be automatically restored. For the SU($N$) group, the integrations (see Eq. (3)) of the fundamental representation are

$$\int \mathrm{d}U\,[U^*]_{\alpha_1}^{\alpha_1'} U_{\beta_1}^{\beta_1'} = \frac{1}{N}\delta^{\alpha_1'\beta_1'}\delta_{\alpha_1\beta_1} \tag{44}$$

and

$$\int dU \, [U^*]_{\alpha_1}^{\alpha'_1} [U^*]_{\alpha_2}^{\alpha'_2} U_{\beta_1}^{\beta'_1} U_{\beta_2}^{\beta'_2} = \frac{1}{N^2-1} \left( \delta^{\alpha'_1 \beta'_1} \delta^{\alpha'_2 \beta'_2} \delta_{\alpha_1 \beta_1} \delta_{\alpha_2 \beta_2} + \delta^{\alpha'_1 \beta'_2} \delta^{\alpha'_2 \beta'_1} \delta_{\alpha_1 \beta_2} \delta_{\alpha_2 \beta_1} \right)$$
$$- \frac{1}{(N^2-1)N} \left( \delta^{\alpha'_1 \beta'_2} \delta^{\alpha'_2 \beta'_1} \delta_{\alpha_1 \beta_1} \delta_{\alpha_2 \beta_2} + \delta^{\alpha'_1 \beta'_1} \delta^{\alpha'_2 \beta'_2} \delta_{\alpha_1 \beta_2} \delta_{\alpha_2 \beta_1} \right). \tag{45}$$

As a consequence, the symmetrized Green's function defined in Eq. (8) is given by

$$\bar{G}_{\alpha_1 \alpha_2}(1,2) = \overline{\langle \psi^*_{\alpha_2}(2) \psi_{\alpha_1}(1) \rangle} = \int dU \, [U^*]_{\alpha_1}^{\alpha'_1} U_{\alpha_2}^{\alpha'_2} \langle \psi^*_{\alpha'_2}(2) \psi_{\alpha'_1}(1) \rangle$$
$$= \frac{1}{2} \delta_{\alpha_1 \alpha_2} \left[ \langle \psi^*_\uparrow(2) \psi_\uparrow(1) \rangle + \langle \psi^*_\downarrow(2) \psi_\downarrow(1) \rangle \right] \tag{46}$$
$$= \frac{1}{2} \delta_{\alpha_1 \alpha_2} \left[ G_{\uparrow\uparrow}(1,2) + G_{\downarrow\downarrow}(1,2) \right].$$

And for the spin-$z$ correlation defined in Eq. (11), the symmetrization gives

$$\chi_{\rm sp}(1,2) \equiv \bar{\chi}_{S^z S^z}(1,2) = \frac{1}{3} \sum_{b=x,y,z} \chi_{S^b S^b}(1,2). \tag{47}$$

Actually, in this case, the translation invariance has been restored after the symmetrization. To clarify this, we use a symmetry of a $z$-polarized pure AF system:

$$\langle \psi^*_{\alpha_1}(\boldsymbol{x}_1) \ldots \psi^*_{\alpha_n}(\boldsymbol{x}_n) \psi_{\beta_1}(\boldsymbol{x}'_1) \ldots \psi_{\beta_m}(\boldsymbol{x}'_m) \rangle$$
$$= \langle i\psi^*_{\bar{\alpha}_1}(\boldsymbol{x}_1 + \boldsymbol{a}_x) \ldots i\psi^*_{\bar{\alpha}_n}(\boldsymbol{x}_n + \boldsymbol{a}_x) i\psi_{\bar{\beta}_1}(\boldsymbol{x}'_1 + \boldsymbol{a}_x) \ldots i\psi_{\bar{\beta}_m}(\boldsymbol{x}'_m + \boldsymbol{a}_x) \rangle, \tag{48}$$

where the $\bar{\alpha}$ means flipping the spin, i.e., $\bar{\alpha} = \downarrow$ when $\alpha = \uparrow$. This symmetry transformation $\psi_\alpha(\boldsymbol{x}) \to i\psi_{\bar{\alpha}}(\boldsymbol{x} + \boldsymbol{a}_x)$ is combination of a rotation of $\pi$ around the $x$-axis and a translation of the smallest unit along the $x$-axis. For the Green's function, it is quite simple that

$$G_{\alpha_1 \alpha_2}(1,2) = G_{\bar{\alpha}_1 \bar{\alpha}_2}(1 + \boldsymbol{a}_x, 2 + \boldsymbol{a}_x). \tag{49}$$

Then one can prove that

$$\bar{G}_{\alpha_1 \alpha_2}(1,2) = \bar{G}_{\alpha_1 \alpha_2}(1 + \boldsymbol{a}_x, 2 + \boldsymbol{a}_x). \tag{50}$$

For the spin correlation, one can first consider how the spin field operator changes. For spin-$z$ operator,

$$S^z(1) = \sum_{\alpha\beta} \psi^*_\alpha(1) \sigma^z_{\alpha\beta} \psi_\beta(1)$$
$$\to \sum_{\alpha\beta} \psi^*_{\bar{\alpha}}(1 + \boldsymbol{a}_x) \sigma^z_{\alpha\beta} \psi_{\bar{\beta}}(1 + \boldsymbol{a}_x) = -S^z(1 + \boldsymbol{a}_x), \tag{51}$$

similarly, $S^x(1) \to S^x(1 + \boldsymbol{a}_x), S^y(1) \to -S^y(1 + \boldsymbol{a}_x)$. As a result, (for $b = x,y,z$),

$$\chi_{S^b S^b}(1,2) = \chi_{S^b S^b}(1 + \boldsymbol{a}_x, 2 + \boldsymbol{a}_x). \tag{52}$$

Thus, it is clear that

$$\bar{\chi}_{S^z S^z}(1,2) = \bar{\chi}_{S^z S^z}(1 + \boldsymbol{a}_x, 2 + \boldsymbol{a}_x). \tag{53}$$

Given that translational invariance has been shown to be restored (refer to Eq. (50) and Eq. (53)), it is now valid to apply the Fourier transform (over the whole 2D square lattice rather than the A-B sublattice) to these symmetrized quantities. This allows us to relate the

2D square reciprocal lattice to the A-B reciprocal lattice and to perform the symmetrization directly in momentum space, which may be convenient in some cases. The trick is to do the following. Through Eq. (50), we can express the Green's function as

$$\bar{G}(x_1, x_2) = \frac{1}{2} \big[ \bar{G}(x_1, x_2) + \bar{G}(x_1 + a_x, x_2 + a_x) \big], \tag{54}$$

where, for convenience, the imaginary time $\tau$ is omitted. Without losing generality, one sets $x_1 = R + u_l$ and $x_2 = 0$, and obtains

$$\bar{G}(R + u_l, 0) = \frac{1}{2} \big[ \bar{G}(R + u_l, 0) + \bar{G}(R + u_l + a_x, a_x) \big]. \tag{55}$$

Substituting $u_A = 0$ and $u_B = a_x$, Eq. (55) yields

$$\bar{G}(R, 0) = \frac{1}{2} \big[ \bar{G}(R, 0) + \bar{G}(R + a_x, a_x) \big], \tag{56}$$

$$\bar{G}(R + a_x, 0) = \frac{1}{2} \big[ \bar{G}(R + a_x, 0) + \bar{G}(R + a_1, a_x) \big]. \tag{57}$$

The Green's functions in the right hand side can be expressed by A-B lattice representation (see Sec.3.4), as

$$\bar{G}(R, 0) = \frac{1}{2} \big[ \bar{G}^{AA}(R, 0) + \bar{G}^{BB}(R, 0) \big], \tag{58}$$

$$\bar{G}(R + a_x, 0) = \frac{1}{2} \big[ \bar{G}^{BA}(R, 0) + \bar{G}^{AB}(R + a_1, 0) \big]. \tag{59}$$

Or namely,

$$\bar{G}(R) = \frac{1}{2} \big[ \bar{G}^{AA}(R) + \bar{G}^{BB}(R) \big], \tag{60}$$

$$\bar{G}(R + a_x) = \frac{1}{2} \big[ \bar{G}^{AB}(R + a_1) + \bar{G}^{BA}(R) \big], \tag{61}$$

where $\bar{G}(x)$ means $\bar{G}(x, 0)$ and $\bar{G}^{l_1 l_2}(R)$ means $\bar{G}^{l_1 l_2}(R, 0)$. Then, the Fourier transformation (on the 2D square lattice) is

$$\begin{aligned}
\tilde{\bar{G}}(k \in \mathcal{B}) &= \sum_x e^{-ik \cdot x} \bar{G}(x) = \sum_R e^{-ik \cdot R} \bar{G}(R) + e^{-ik \cdot a_x} \sum_R e^{-ik \cdot R} \bar{G}(R + a_x) \\
&= \frac{1}{2} \sum_R e^{-ik \cdot R} \big[ \bar{G}^{AA}(R) + \bar{G}^{BB}(R) + e^{ik \cdot a_x} \bar{G}^{AB}(R) + e^{-ik \cdot a_x} \bar{G}^{BA}(R) \big].
\end{aligned} \tag{62}$$

Denote the momentum space of the A-B sublattice by $\mathcal{B}_{AB}$. If $k \in \mathcal{B}_{AB}$, the right hand side of the Eq. (62) is just the Fourier transform in the A-B sublattice defined as Eq. (37). One can divide the $\mathcal{B}$ into $\mathcal{B}_{AB}$ and $\{k + Q | k \in \mathcal{B}_{AB}\}$. Then, since $Q \cdot (n_1 a_1 + n_2 a_2) = 2\pi(n_1 + n_2)$ and $Q \cdot a_x = \pi$, for $k \in \mathcal{B}_{AB}$ one has

$$\tilde{\bar{G}}(k) = \frac{1}{2} \big[ \tilde{\bar{G}}^{AA}(k) + \tilde{\bar{G}}^{BB}(k) + e^{ik \cdot a_x} \tilde{\bar{G}}^{AB}(k) + e^{-ik \cdot a_x} \tilde{\bar{G}}^{BA}(k) \big], \tag{63}$$

$$\tilde{\bar{G}}(k + Q) = \frac{1}{2} \big[ \tilde{\bar{G}}^{AA}(k) + \tilde{\bar{G}}^{BB}(k) - e^{ik \cdot a_x} \tilde{\bar{G}}^{AB}(k) - e^{-ik \cdot a_x} \tilde{\bar{G}}^{BA}(k) \big]. \tag{64}$$

Similar formula also holds for two-body correlation functions. Substituting

$$\tilde{\bar{G}}^{l_1 l_2}(k) = \frac{1}{2} \sigma^0 \big[ \tilde{G}^{l_1 l_2}_{\uparrow\uparrow}(k) + \tilde{G}^{l_1 l_2}_{\downarrow\downarrow}(k) \big], \tag{65}$$

which is the Eq. (46) in the A-B sublattice reciprocal space, into the Eqs. (63, 64), we accomplish the symmetrization directly in momentum space. And one can also go back to the position space directly from the Eqs. (63, 64) by

$$\bar{G}(\mathbf{R}) = \frac{1}{2N} \sum_{\mathbf{k} \in \mathcal{B}_1} e^{i\mathbf{k} \cdot \mathbf{R}} \left[ \bar{\tilde{G}}(\mathbf{k}) + \bar{\tilde{G}}(\mathbf{k} + \mathbf{Q}) \right], \tag{66}$$

$$\bar{G}(\mathbf{R} + \mathbf{a}_x) = \frac{1}{2N} \sum_{\mathbf{k} \in \mathcal{B}_1} e^{i\mathbf{k} \cdot \mathbf{R}} e^{i\mathbf{k} \cdot \mathbf{a}_x} \left[ \bar{\tilde{G}}(\mathbf{k}) - \bar{\tilde{G}}(\mathbf{k} + \mathbf{Q}) \right], \tag{67}$$

where $2N$ is number of lattice points of the 2D square lattice. Regarding the correlation function, an analogous methodology is employed, the details of which are omitted herein for conciseness.

## 4 Results

We investigated systems with intermediate-to-strong interactions at half filling, benchmarking our results against DQMC. To ensure the accuracy of DQMC results, we test the Trotter error of momentum distribution and charge correlation down to $\beta = 20$ for $U = 8$ system (see Appendix C). We focus on spin correlation with strong coupling $U = 8$ at temperatures down to $\beta = 16$, the corresponding Green's function results are also presented. Given the huge computational cost of DQMC, this benchmarking is conducted on $12 \times 12$ lattices. Additionally, we present benchmark of spin correlation for $\beta = 8$ and $0 \le U \le 8$ on $16 \times 16$ lattice, demonstrating the reliability of the symmetrization scheme across different interaction strengths and larger system sizes. We also investigated the spin correlation and the Green's function with intermediate coupling $U = 4$ at temperatures down to $\beta = 20$, but the results are presented in Appendix D rather than in this section.

### 4.1 Spin correlation function

We are concerned with the spin correlation function $\chi_{\mathrm{sp}}(1, 2) \equiv \langle S^z(1) S^z(2) \rangle$. In the paramagnetic phase, $\chi_{\mathrm{sp}} = \chi_{S^z S^z}$, where $\chi_{S^a S^b}$ defined as Eq. (11) with $a, b = x, y, z$. In the pseudo AF phase, in contrast, the contribution of the AF order parameter should be considered separately, refer to Ref. [69]. Thus the spin correlation function becomes

$$\tilde{\chi}_{\mathrm{sp}}(i\omega_n, \mathbf{q}) = \delta_{\mathbf{q}, \mathbf{Q}} \beta N_{\mathrm{sq}} |S_0^z|^2 + (1 - \delta_{\mathbf{q}, \mathbf{Q}}) \tilde{\chi}_{S^z S^z}(i\omega_n, \mathbf{q}), \tag{68}$$

where $N_{\mathrm{sq}}$ is number of the points of the 2D square lattice, and the $S_0^z$ is the AF order (as shown in Eq. (43)). Applying the symmetrization scheme, we derive the formula to calculate spin correlation function in the pseudo AF phase, as

$$\bar{\tilde{\chi}}_{\mathrm{sp}}(i\omega_n, \mathbf{q}) = \delta_{\mathbf{q}, \mathbf{Q}} \frac{\beta N_{\mathrm{sq}}}{3} |S_0^z|^2 + (1 - \delta_{\mathbf{q}, \mathbf{Q}}) \bar{\tilde{\chi}}_{S^z S^z}(i\omega_n, \mathbf{q}). \tag{69}$$

Based on the methodology in Sec. 3.5, the symmetrized spin fluctuation should be written as $\bar{\chi}_{S^z S^z} = \frac{1}{3}(\chi_{S^x S^x} + \chi_{S^y S^y} + \chi_{S^z S^z})$, and we obtain the spin correlation function in the reciprocal 2D square lattice by that in the reciprocal A-B lattice using

$$\bar{\tilde{\chi}}_{S^z S^z}(i\omega_n, \mathbf{k}) = \frac{1}{2} \left[ \begin{array}{l} \bar{\tilde{\chi}}_{S^z S^z}^{AA}(i\omega_n, \mathbf{k}) + e^{i\mathbf{k} \cdot \mathbf{a}_x} \bar{\tilde{\chi}}_{S^z S^z}^{AB}(i\omega_n, \mathbf{k}) \\ + \bar{\tilde{\chi}}_{S^z S^z}^{BB}(i\omega_n, \mathbf{k}) + e^{-i\mathbf{k} \cdot \mathbf{a}_x} \bar{\tilde{\chi}}_{S^z S^z}^{BA}(i\omega_n, \mathbf{k}) \end{array} \right], \tag{70}$$

$$\bar{\tilde{\chi}}_{S^z S^z}(i\omega_n, \mathbf{k} + \mathbf{Q}) = \frac{1}{2} \left[ \begin{array}{l} \bar{\tilde{\chi}}_{S^z S^z}^{AA}(i\omega_n, \mathbf{k}) - e^{i\mathbf{k} \cdot \mathbf{a}_x} \bar{\tilde{\chi}}_{S^z S^z}^{AB}(i\omega_n, \mathbf{k}) \\ + \bar{\tilde{\chi}}_{S^z S^z}^{BB}(i\omega_n, \mathbf{k}) - e^{-i\mathbf{k} \cdot \mathbf{a}_x} \bar{\tilde{\chi}}_{S^z S^z}^{BA}(i\omega_n, \mathbf{k}) \end{array} \right]. \tag{71}$$

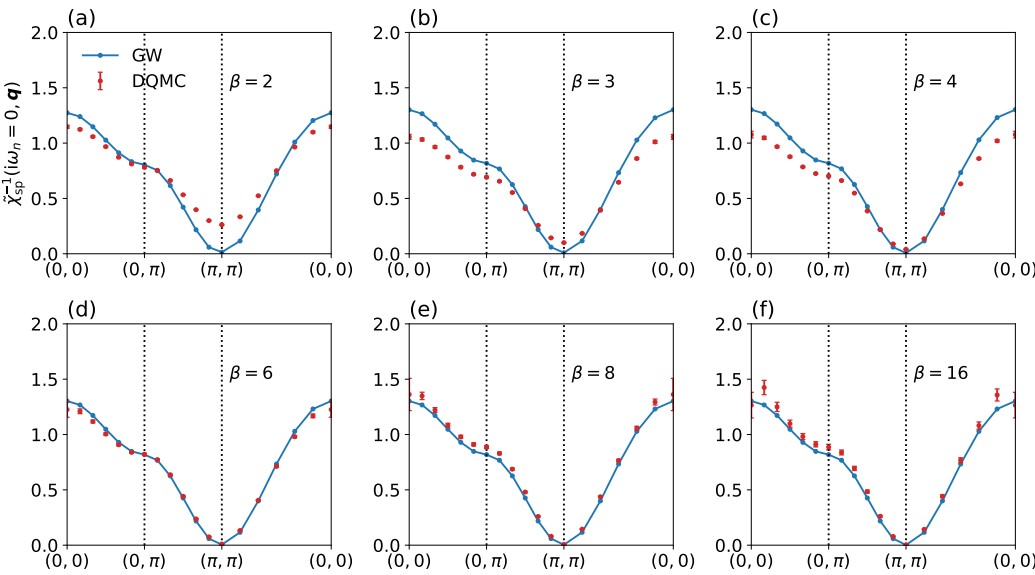

Figure 2: Momentum dependence of the inverse spin susceptibility for the half-filled Hubbard model on a $12 \times 12$ lattice with $U = 8$. Results are shown for decreasing temperatures, corresponding to inverse temperatures (a) $\beta = 2$, (b) $\beta = 3$, (c) $\beta = 4$, (d) $\beta = 6$, (e) $\beta = 8$, and (f) $\beta = 16$. The momentum path follows $(0,0) \rightarrow (0,\pi) \rightarrow (\pi,\pi) \rightarrow (0,0)$. Blue curves denote the symmetrized GW-covariance approximation, while red symbols with error bars show DQMC benchmarks. A systematic improvement in the agreement between GW-covariance and DQMC is observed as temperature is lowered.

The symmetrized GW-covariance and DQMC numerical results in $12 \times 12$ lattice for the spin correlation function $\tilde{\chi}_{sp}(i\omega = 0, \boldsymbol{q})$ with $U = 8$ at half filling are presented in Fig. 2. Qualitatively, the GW-covariance results are consistent with DQMC, exhibiting extrema at the AF wave vector $\boldsymbol{Q} = \pi\boldsymbol{b}_x + \pi\boldsymbol{b}_y$ and the zero momentum $\boldsymbol{k} = \boldsymbol{0}$, and relatively small slope at the anti-nodal point $\boldsymbol{k}^{AN} = \pi\boldsymbol{b}_y$. The AF solution of the GW equations emerges at about pseudo critical (crossover) temperature $T_c \simeq 0.60$, or $\beta_c \simeq 1.68$. Near the critical temperature $\beta_c$, the GW-covariance results show relatively poor agreement with the DQMC results. As the inverse temperature $\beta$ increases far away from $\beta_c$, quantitatively, the our results gradually show more close agreement with DQMC. At low temperatures far from the critical region ($\beta \gtrsim 6$), the GW-covariance results are in good agreement with the DQMC benchmarks.

As presented in Fig. 3, we calculate the short-range behavior of the correlation function. Qualitatively the GW-covariance results are basically consistent with the DQMC results, but quantitatively the absolute values of GW-covariance are larger. This is consistent with our experience that GW tends to overestimate the influence of magnetic order on the system. To solve the problem, we introduce an estimation using the $\chi$-sum rule. The $\chi$-sum rule, i.e.,

$$\chi_{ch}(\tau = 0, \boldsymbol{r} = \boldsymbol{0}) + \chi_{sp}(\tau = 0, \boldsymbol{r} = \boldsymbol{0}) = 2\rho - \rho^2 \tag{72}$$

relates the charge correlation function $\chi_{ch}$ and spin correlation function $\chi_{sp}$ to the charge density $\rho$. It is a direct consequence of the Pauli exclusion principle, refer to Appendix E for its derivation. The $\chi$-sum rule provides an estimate for $\tilde{\tilde{\chi}}_{sp}^{(sr)}(i\omega_n = 0, \boldsymbol{Q})$, as given by

$$\tilde{\tilde{\chi}}_{sp}^{(sr)}(i\omega_n = 0, \boldsymbol{Q}) = \beta N_{sq}(2\rho - \rho^2) - \sum_{i\omega_n \neq 0} \sum_{\boldsymbol{q} \neq \boldsymbol{Q}} \tilde{\tilde{\chi}}_{sp}(i\omega_n, \boldsymbol{q}) - \sum_{i\omega_n, \boldsymbol{q}} \tilde{\tilde{\chi}}_{ch}(i\omega_n, \boldsymbol{q}), \tag{73}$$

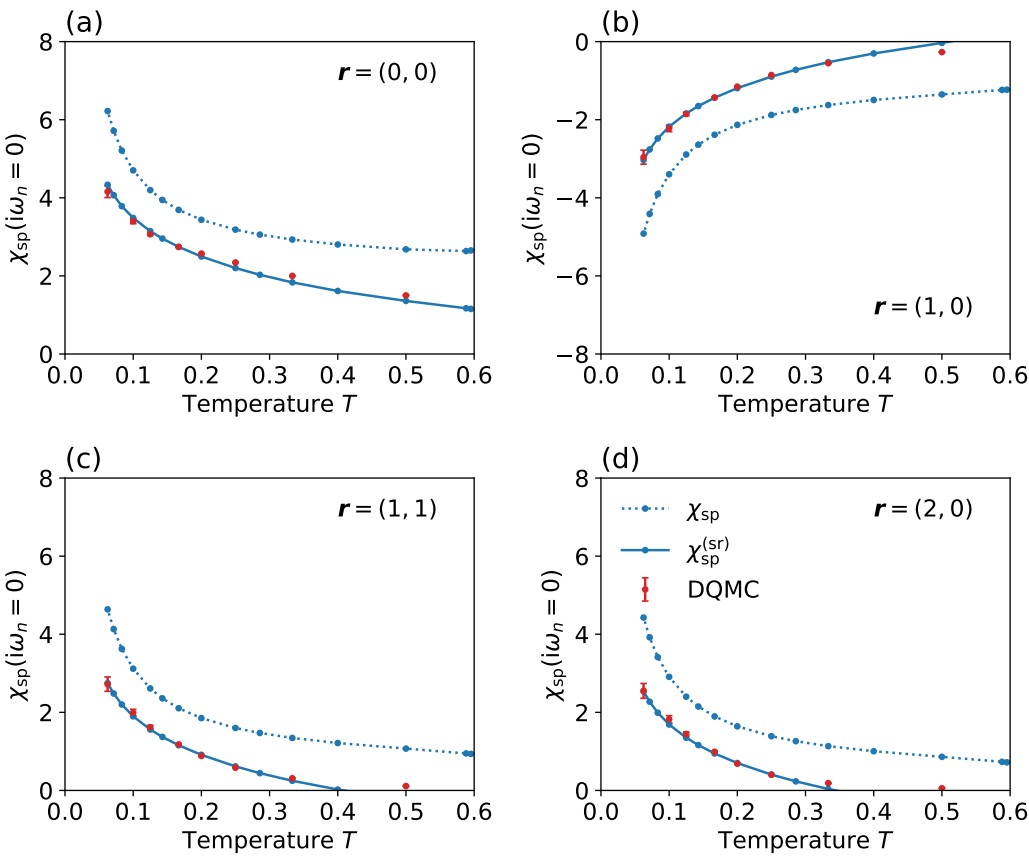

Figure 3: Temperature dependence of the static spin correlation function in spatial space for the half-filled Hubbard model ($U = 8$) on a $12 \times 12$ lattice. Panels (a)-(d) display results for lattice separations $\mathbf{r} = (0,0)$, $(1,0)$, $(1,1)$, and $(2,0)$, respectively. The DQMC benchmarks (red points with error bars) are compared with the GW-covariance result $\chi_{\mathrm{sp}}$ (blue dashed curve) and the $\chi$-sum rule estimate $\chi_{\mathrm{sp}}^{(\mathrm{sr})}$ (blue solid curve).

as long as the approximate correlations used on the right-hand side are reliable. The results are also are presented in Fig. 3. The $\bar{\chi}_{\mathrm{sp}}^{(\mathrm{sr})}$ agrees very well with the DQMC at low temperature away from the pseudo critical $T_c$, i.e., at $T \leq 0.25$. Near $T_c$, the short-range component of $\bar{\chi}_{\mathrm{sp}}^{(\mathrm{sr})}$ exhibits better behavior than its longer-range counterpart. Specifically, the $T$-$\chi$ curves cross zero at $\mathbf{r} = \mathbf{a}_x$, $\mathbf{a}_x + \mathbf{a}_y$, and $2\mathbf{a}_x$ for $T > 0.5$, $0.4$, and $0.3$, respectively. The correlation length, as calculated by our approach in the pseudo AF state, is overestimated near the critical region, which coincides with the poor behavior of the momentum-space results in this temperature region shown in Fig. 2. Consequently, in this temperature region, the longer-range component is less reliable.

We then turn to consider the influence of different interaction strengths $U$. As shown in Fig. 4, in the $16 \times 16$ system, the spin correlation function is calculated with $\chi_{\mathrm{sp}}(i\omega_n = 0, \mathbf{r} = \mathbf{0})$ as a representative. The temperature is fixed at $\beta = 8$, and the interaction strength $U$ gradually increases from zero to the typical strong correlation region $U = 8$. When the interaction $U < U_c$ ($U_c \sim 2.8$), the GW predicts paramagnetic phases. We identify it as the Slater branch. When the interaction is relatively weak ($U \lesssim 2.5$), GW-covariance closely agrees with DQMC. However, when $2.5 \lesssim U \lesssim U_c$, affected by the AF instability $\chi_{\mathrm{sp}}(i\omega_n = 0, \mathbf{Q}) \to \infty$, there is a serious overestimation for GW-covariance. When $U > U_c$, the GW predicts AF phases. We

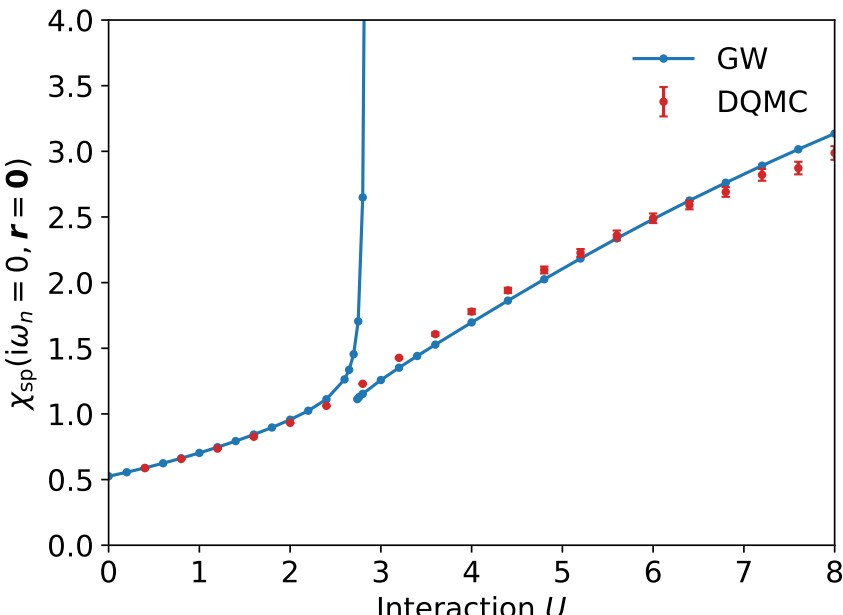

Figure 4:   Spin correlation function as a function of interaction strength $U$ for a $16 \times 16$ lattice at low temperature ($\beta = 8$). The DQMC results (red discrete points with error bars) exhibit a continuous evolution. In contrast, the GW-covariance approximation (blue curve) reveals a two-branch structure: a paramagnetic (Slater) branch for $U < U_c$ and a pseudo-antiferromagnetic (Mott-Heisenberg) branch for $U > U_c$, with a critical $U_c \simeq 2.8$ leading to a pronounced discontinuity. The GW-covariance results on the antiferromagnetic branch are obtained via the $\chi$-sum rule estimate.

identify it as the Mott-Heisenberg branch. Within the range we considered, the deviation from DQMC is not significant, indicating that the symmetrized GW-covariance theory performs well. This sheds light on the calculation of strongly correlated systems.

## 4.2   Green's function

In this section, we calculate the Green's function at low temperature using the symmetrization process introduced in Sec. 3.5. We are concerned with the nodal point $\boldsymbol{k}^N = (\pi/2)(\boldsymbol{b}_x + \boldsymbol{b}_y)$ and the anti-nodal point $\boldsymbol{k}^{AN} = \pi \boldsymbol{b}_x$ on the Fermi surface. The benchmark is depicted in Fig. 5, within the spurious AF region. As the temperature decreases from $\beta = 2$ near critical temperature to deep AF region, the symmetrized Green's function calculated by GW shows an increasingly closer agreement with the results obtained from DQMC. This tendency holds until extremely low temperature $\beta = 16$, which we do not show here. Specifically, we are concerned with the equal-time Green's function $G^{N/AN}(\tau = 0)$ near and the minimal value of the Green's function $G^{N/AN}(\tau = \beta/2)$ near the Fermi surface. $G^{N/AN}(\tau = 0)$ corresponds to the momentum distribution of the density, which is crucial for understanding the electronic structure of the system [58]. On the other hand, $G^{N/AN}(\tau = \beta/2)$ can serve as a proxy for spectral function $A^{N/AN}(\omega = 0) \simeq \frac{\beta}{2} G^{N/AN}(\tau = \beta/2)$, helping us to probe the emergence of the pseudogap [70]. We find all of them become close to the DQMC results, demonstrating the effectiveness of the symmetrized GW method at extremely low temperatures.

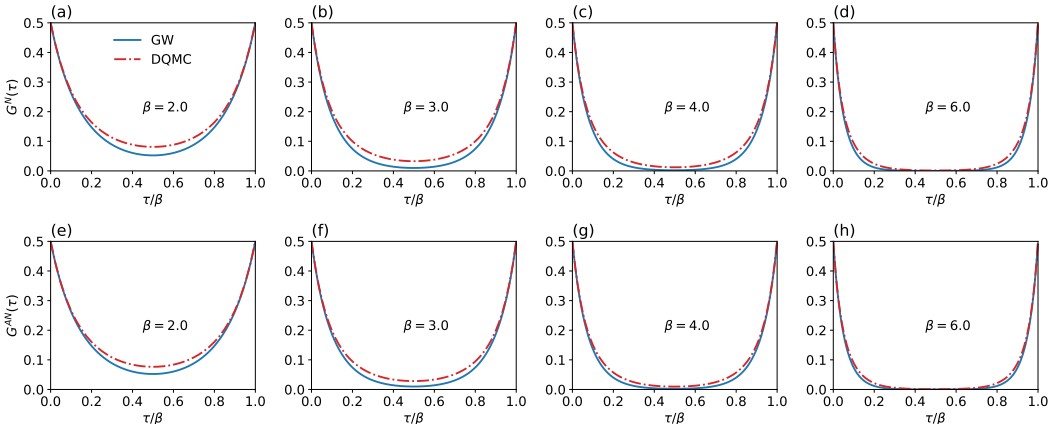

Figure 5: Imaginary-time Green's function $G(\mathbf{k}, \tau)$ for the half-filled Hubbard model on a $12 \times 12$ lattice with $U = 8$, comparing the symmetrized GW approximation (blue solid lines) and DQMC results (red dot-dashed lines). Panels (a)-(d) correspond to the nodal point $\mathbf{k} = (\pi/2, \pi/2)$, while panels (e)-(h) show the antinodal point $\mathbf{k} = (\pi, 0)$. The columns represent increasing inverse temperatures, i.e., (a,e) $\beta = 2$, (b,f) $\beta = 3$, (c,g) $\beta = 4$, and (d,h) $\beta = 6$.

## 5 The criterion for effectiveness of approximation method

After benchmarking the symmetrization GW-covariance approximation with DQMC at half-filling, we expect to use the approximation to investigate doped regions of the Hubbard model, which exhibit various physical phenomena including CDW and superconducting states [1–5]. These systems are challenging for existing unbiased numerical methods. For example, the DQMC suffers from the fermion sign problem and struggles to converge at low temperatures [20–24]. The DMRG is typically limited to cylindrical geometries, and its computational cost grows exponentially with sizes of 2D quantum systems, restricting it to very small system sizes [71–73]. As a result, it is difficult to reliably benchmark many-body approximation methods in doped regions. While the symmetrized GW-covariance method already performs well in benchmarks at half-filling, further evidence would strengthen the case for its reliability in doped regimes. Therefore, it would be beneficial to have a self-consistent criterion for assessing reliability, one that is independent of third-party data and based solely on the results of the approximation itself.

A natural way to evaluate the effectiveness of an approximation is to examine the fundamental physical relations such as the conservation laws. There are three fundamental relations, i.e., the FDR, the WTI and local momentum sum rules based on the Pauli exclusion principle. The FDR provides an essential bridge between theory and experiment by relating fundamental two-body correlation functions to experimentally measurable response functions. This is exemplified by the Kubo formula, which expresses electrical conductivity in terms of the current-current correlation function. The WTI stems from the invariance of a system under a symmetry transformation, and reflects the conservation law corresponding to the symmetry, and is therefore important for the consistency checking of various theoretical methods. For instance, the WTI corresponding to the U(1) gauge invariance reflects the current conservation law. Furthermore, the $f$-sum rule, an implication of the WTI, is also used in experimental analysis. Therefore, both the FDR and the WTI are crucial. Given that covariance theory [59] now enables approaches that respect these two laws, we will now discuss the third relation.

It has been conjectured in Ref. [3] that no theoretical approach for correlated electrons,

except for the exact solution, should be able to fulfill both of the two requirements: (i) all conservation laws and the related sum rules, and (ii) sum rules for one- and two-particle Green's functions based on the Pauli principle. Assuming that the conjecture holds, once the FDR and the WTI are satisfied, the Pauli principle becomes a straightforward criterion for assessing the reliability of an approximation method. Specifically, a smaller deviation in the local momentum sum rules (based on the Pauli principle) for one- and two-particle Green's functions generally indicates a closer agreement with the exact theory.

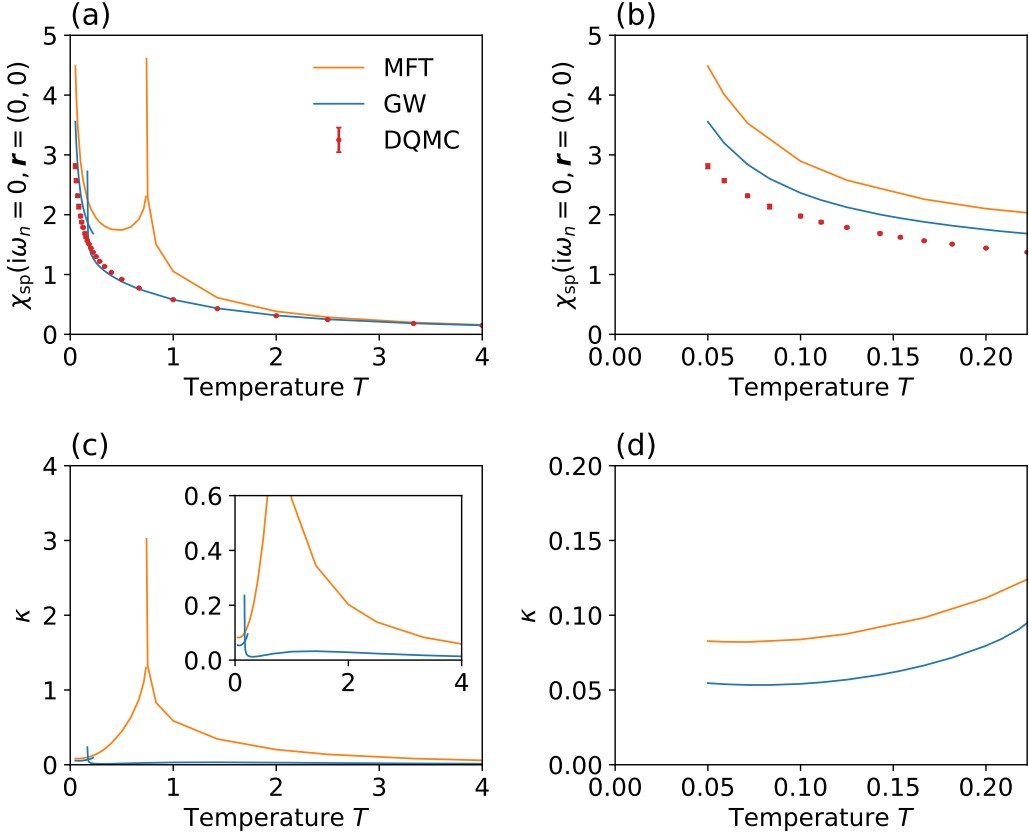

Figure 6: (a) Temperature dependence of the static spin susceptibility $\chi_{sp}$, showing results from mean-field-covariance (orange, PM/AF branches), GW-covariance (blue, PM/AF branches), and DQMC (red points). (b) Expanded view of the low-temperature range. (c) Corresponding measure of the Pauli principle violation $\kappa$ over the full temperature range; the inset magnifies the small-$\kappa$ region. (d) The $\kappa$ values in the low-temperature range shown in (b).

To verify the criterion, we compare the $\chi$-sum rule deviations of the symmetrized mean-field-covariance (MF-covariance) with those of the symmetrized GW approximation. In previous sections, we use symmetrization scheme and the covariance theory based on the GW approximation to approach the correlation functions in the pseudo AF phase, respecting the FDR and the WTI. The same framework can also be applied to mean-field theory to obtain correlation functions that obey the FDR and the WTI. Indeed, due to the simplicity of the mean-field equation, this MF-covariance approach is equivalent to the RPA. Figure 6a presents the two branches (AF and paramagnetic) of the spin correlation functions calculated using MF-covariance, GW-covariance, and DQMC. The deviations from the $\chi$-sum rule, quantified by the parameter $\kappa$ (see Eq. 115), are shown in Fig. 6c. For clarity, Fig. 6b and Fig. 6d show enlarged

views of the respective subfigures. Both the GW-covariance and MF-covariance results agree well with the DQMC benchmarks at extremely high and low temperatures that are far from the crossover region (i.e., near the pseudo AF critical point), and the corresponding $\kappa$ values are small ($< 10\%$ for GW-covariance and $< 20\%$ for MF-covariance) at these temperatures. However, in the crossover region, the MFT results deviate significantly from the DQMC benchmarks, and their $\kappa$ values go up to $> 100\%$. In contrast, the GW-covariance results in this region are much more reliable than those of MF-covariance. The corresponding $\kappa$ values for the AF and paramagnetic branches remain below 10% and 5%, respectively, except at points extremely close to the instability in the paramagnetic branch. These results generally support our conjecture that a smaller deviation from the local momentum sum rules indicates greater reliability.

## 6 Conclusion and Discussion

### 6.1 Summary

In this work, we establish a general symmetrization scheme and test its effectiveness in the 2D Hubbard model within the GW and GW-covariance approximation. In the pseudo AF phase predicted by the GW approximation, we calculate the single-particle Green's function and the spin correlation function. By comparing with the DQMC method at strong coupling $U = 8$ and intermediate coupling $U = 4$, we confirm the scheme's effectiveness at sufficiently low temperatures. Moreover, we calculate the correlation functions at $\beta = 8$ for a range of interactions up to $U = 8$, including the paramagnetic and pseudo AF phases. These results demonstrate the reliability of our method. Regarding the fundamental relations, the covariance theory ensures the the satisfaction of the FDR and WTI, while our numerical results indicate that at low temperatures far from the crossover region, deviations from the $\chi$-sum rule are below 10% for $U = 4$ and below 20% for $U = 8$. We further conjecture a self-consistency criterion for many-body approximation methods: provided that the FDR and WTI are satisfied, the reliability of a method is determined by its degree of violation of the Pauli exclusion principle. The degree of violation of the Pauli exclusion principle can be evaluated by deviations from the local momentum sum rules, such as the $\chi$-sum rule. Our work offers a novel framework for exploring the rich physics of the 2D Hubbard model in the strong-coupling regime ($U \gtrsim 8$), which is relevant to real high-$T_c$ cuprate superconductors.

### 6.2 Outlook

From the work and results presented in the article, we can draw the following insights. Even in the absence of continuous symmetry breaking in a 2D system, it is possible to select a group-symmetry-broken solution as the starting point for calculations. This solution occurs at a suitably low temperature that is below the instability temperature of the group-invariant phase and hence is inaccessible from the group-invariant solution itself. The example demonstrated here is that an AF instability temperature exists in a paramagnetic phase at half-filling for any positive $U$. At temperatures below this, one should start from the AF solution and apply the symmetrization scheme, even though no spontaneous SU(2)-symmetry-breaking solutions can physically persist for finite lattices or infinite lattices in dimensions lower or equal to two. This symmetrized approach is not in contradiction with, but rather consistent with the Mermin-Wagner theorem. After the symmetrization, any symmetry-broken physical quantities will become symmetry-invariant once again. This coincides with the restoration of symmetry due to strong interactions of Goldstone modes in low dimensions. Specifically, Goldstone modes in the Heisenberg universality class in 2D acquire a small mass due to the interaction, and

long-range order is consequently destroyed.

The symmetrized physical quantities obtained through this procedure remain reliable, as demonstrated by the physical quantities presented in this paper. Even for order parameter field correlations, the approach can still provide a fairly reliable estimation of short-range correlations. At sufficiently low temperatures, where the correlation length becomes large, the method yields quite good approximations for spin-spin correlations even at few lattice constant apart. Although we primarily demonstrate parameter field correlations (spin-spin correlations) in this work, using this approach we also accurately calculate other physical quantities, such as the density-density correlations shown in the appendix. Based on the evidence presented above, we speculate that, except for order parameter field (in this case, spin field) correlations, the method can also accurately compute other two-body correlations, such as density-density, superconducting pairing field, or current-current correlations, provided the system is not too close to critical or pseudo-critical phase transition lines. Hence, our approach is promising for studying high-$T_c$ superconductivity within the Hubbard model.

The mean-field results do not yield superconductivity. However, detailed studies in Refs. [74, 75] have indicated that spin and density orders emerge at different doping levels, forming a rich variety of phases (such as spiral, stripes, and beat states). According to Refs. [74, 75], fluctuations of the Goldstone modes will restore the spin SU(2) symmetry for the 2D Hubbard model. For approaches beyond mean-field theory, such as GW, GW-EDMFT, and TRILEX [76], although the superconducting instability appears at $T_c$, when cooling gradually from a sufficiently high temperature in the underdoped regime, the AF instability or other magnetic and density wave instabilities appear earlier at $T_{AF} > T_c$, thereby preventing the system from directly reaching $T_c$ [76]. However, within our symmetrization scheme, the SU(2)-symmetrized superconducting instability can be calculated in the pseudo AF phase (or pseudogap phase), as well as in spiral or stripe phases with other spin and density orders, thereby enabling the determination of $T_c$. In the overdoped regime, in contrast, no such magnetic or density wave instabilities are expected, and the only instability that arises as the temperature is lowered is the superconducting instability.

The phase transitions in 2D systems with O($N$) symmetry are discussed, for example, in David Tong's lecture [77]. When $N = 1$, it corresponds to the Ising type with discrete $Z_2$ symmetry, where phase transitions caused by discrete symmetry breaking are permitted. When $N = 2$, the Berezinskii-Kosterlitz-Thouless phase transition occurs, which is a topological phase transition without spontaneous continuous symmetry breaking. When $N \geq 3$, there is no phase transition at finite temperature, but only a crossover occurs between the high-temperature paramagnetic phase and the fluctuating AF phase, neither of which exhibits long-range order. Notably, the magnetism of universality class for 2D Hubbard model at half filling corresponds to the case of $N = 3$. It should be noted that a quantum phase transition occurs in 2D systems only at zero temperature for $N = 3$. Since low-temperature phenomena are closer to those at zero temperature, it is not surprising that a crossover exists between the low-temperature regime and the "high-temperature phase". Although no true long-range order exists at finite temperatures, the correlation function is very large at sufficiently low temperatures [31]. Therefore, it is reasonably justified to use the long-range ordered (AF) phase as the starting point of approximation, while the symmetrization scheme should be adopted to ensure consistency with physical reality (i.e., the Mermin-Wagner theorem).

Despite the existence of continuous symmetry breaking in 3D systems at the thermodynamic limit, the symmetrization scheme proposed in this paper remains necessary for finite-sized systems at low temperatures, even in high spatial dimensions. In studies of the 3D Hubbard model, many investigations start with the AF phase and yield physically reasonable results in the thermodynamic limit [28, 78]. However, for finite lattice systems, the symmetrization scheme must be applied, and only the symmetrized results can be compared with Monte Carlo

(MC) results from finite systems, since symmetry breaking is absent and the order parameter field is zero in any finite system. The symmetry breaking occurs only in the thermodynamic limit. Nevertheless, finite-size scaling can be used to identify phase transitions in the thermodynamic limit and to determine whether long-range order exists, using two-body correlation functions of the order parameter field computed for finite lattices within many-body methods. Since the order parameter field is always zero for finite lattices at any temperature, single-body averages of the order parameter field cannot be used to study the long-range order or spontaneous symmetry breaking in the thermodynamic limit from the finite $N$ analysis.

Numerous methods have been attempted to search for the superconducting phase in the low-temperature Hubbard model, and huge progress has been made in recent years. Below, we provide a brief discussion of some recent developments. Note that, as this is not a review paper, we do not intend to present a comprehensive overview of recent advances; accordingly, the references quoted below constitute a selective and inevitably somewhat biased sample of recent works.

A substantial reduction in temperature has been achieved both with and without doping in cold-atom quantum simulations of the Hubbard model, as reported in Ref. [79]. This progress allows access to interesting phases that had not previously been explored in cold-atom quantum simulators, such as spin or charge density wave states emerging at sufficiently low temperatures. However, the temperature is still slightly high, with the maximum inverse temperature is only around $\beta = 10$ [79]. In contrast, the superconducting phase transition could occurs at $\beta \gtrsim 30$.

Huge progress has been made in the last four to five years in numerical studies of the Hubbard and $t-J$ model relevant to high-$T_c$ superconductors. The ground-state phase diagram of the $t-t'-J$ model has been studied using DMRG [71–73] calculations on cylinders of widths 6 and 8 in Refs. [80–82]. These works report coexisting uniform AF and $d$-wave singlet pairing phases at low electron doping, while in the corresponding hole-doped region such behavior does not exist, which is inconsistent with experimental results. However, upon increasing the cylinder width from 4 to 8, a significant strengthening of quasi-long-range superconducting correlations was observed in the hole-doped region [83]. Furthermore, using DMRG with substantially increased bond dimensions, the $d$-wave superconducting state also emerges on the hole-doped side at the optimal 1/8 doping in eight-leg systems of the $t-t'-J$ model [84]. The tangent space tensor renormalization group (tanTRG) method was applied to study the $t-t'-J$ model at finite temperature on cylinders up to width $W = 6$ in Ref. [85], yielding the finite-temperature phase diagram of the model. A clear finite-temperature phase transition to the $d$-wave superconducting state was identified in the electron-doped region, consistent with previous ground-state results for the $t-t'-J$ model. Subsequently, tanTRG was applied to the Hubbard model without a $t'$ term to study at temperatures as low as 1/24 (in unit of the hopping energy) [86]. However, no superconducting phase transition was found. It would be of great interest for the authors of Ref. [86] to investigate the Hubbard model with a $t'$ term to examine whether the finite-temperature superconducting phase transition occurs. The ground state of the Hubbard model with a $t'$ term was studied on long six-leg square cylinders in Ref. [87], where the $d$-wave superconducting phase was found only in the electron-doped region, consistent with results for the $t-t'-J$ model. In Ref. [5], the ground state of the doped $t-t'-U$ Hubbard model on the square lattice was studied using a combination of DMRG and constrained path auxiliary field quantum Monte Carlo on width-4 and width-6 cylinders under the application of a finite global $d$-wave pairing field. The work claims the presence of superconducting phases in both electron- and hole-doped regimes of the two-dimensional Hubbard model with next-nearest-neighbor hopping (i.e., with a $t'$ term), consistent with experimental results for high-$T_c$ superconductivity.

DMRG and tanTRG can be used to study the Hubbard model at zero and finite temperature

without a sign problem in contrast to DQMC. However, the maximum cylinder width currently achievable with these methods is 8, and finite-size effects may still be significant for this width. For finite-temperature studies of the Hubbard model, tanTRG can reach temperatures as low as 1/24 in unit of the hopping energy. In the cuprates, however, the typical hopping energy is approximately 0.4eV, while the superconducting transition temperature at optimal hole doping ranges from 40K to 135K. Expressed in units of the hopping energy, this corresponds to a transition temperature ranging from approximately 1/100 to 1/32. Even the highest transition temperature remains relatively low compared to the hopping energy. This temperature range (from 1/100 to 1/32) represents an intermediate regime that is notoriously challenging for DMRG-liked methods. It remains challenging for DMRG or tanTRG to access even the highest transition temperature.

Despite the progress made with the numerical methods mentioned above, considerable challenges remain in solving the Hubbard model to understand the microscopic mechanism of cuprate high-$T_c$ superconductivity. We are exploring and pursuing a different path to investigate these problems, namely, a non-perturbative many-body method. The method has passed the benchmark test against DQMC results for the Hubbard model without a $t'$ term at half-filling and very low temperatures. Using orthogonal function expansion algorithms, many-body calculations converge exponentially as the number of imaginary time samples increases [88–91], allowing us to access larger lattices at extremely low temperatures while keeping computational costs acceptable. For example, calculations of the GW approximation are feasible for $32 \times 32$ and even $64 \times 64$ lattices at $T = 1/100$. We will use this method to study realistic models with the $t'$ term and away from half-filling (i.e., the doped Hubbard model).

We remind that the GW-covariance approximation is a non-perturbative and analytical method, which still falls within the framework of standard many-body theories such as HF and parquet approximations. The $t - J$ model is a low-energy effective model derived from the Hubbard model in the strong repulsive interaction limit, where doubly occupied sites are projected out. Since there is a double occupancy constraint, most many-body theories handle this via so-called slave-particle formalisms by "fractionalizing" the electron. Many-body theories for the $t - J$ model are not standard in the usual sense, as the constraint requires the introduction of new particles. In contrast to standard many-body theories, it remains unclear how to implement the WTI and FDR, or how to verify the local moment sum rule within such theories. Nevertheless, results from these approaches have been successfully used to explain experimental observations in cuprate superconductors. For example, Refs. [92,93] developed a theory based on the $t - J$ model that explains experimental resistivity data and the relation between the linear slope and the superconducting transition temperature. In this paper, however, we concentrate on the original Hubbard model rather than its low-energy effective strong-coupling version, thereby allowing the use of standard many-body approaches to tackle those problems related to high-$T_c$ cuprate superconductors.

We have applied many-body methods to study the doped Hubbard model. In Ref. [94], we used a PHF method to calculate the Hall coefficient at low temperatures and different doping levels using commonly adopted realistic parameters for the Hubbard model The results are consistent with experimental data from both electron- and hole-doped cuprate materials. In Ref. [69], we applied the GW-covariance method to study the negative-$U$ Hubbard model at quarter filling $n = 0.5$. The Bose-Einstein phase transition temperature was obtained for various $U$, and the Green's functions were calculated and compared with DQMC results, where no fermion sign problem occurs for negative $U$. Thus, the GW-covariance method could be benchmarked against DQMC, and the results for the transition temperature and Green's functions were found to be consistent with DQMC up to intermediate values of $U$. The pseudogap regime was also identified and investigated using a so-called post-GW approach. In particular,

the Goldstone mode, predicted by the WTI in the U(1) symmetry-breaking phase, was clearly identified, demonstrating that the theory is fully consistent with the constraints imposed by conservation laws.

We conclude this paper by quoting the question posed in Ref. [5]: Is there any simple analytic theory of cuprate superconductivity in the style of Bardeen-Cooper-Schrieffer (BCS), or must we always resort to simulation? In our view, the answer to this bold question is: an analytic many-body theory for cuprate superconductivity does exist, but numerical simulations are still essential to validate the correctness of the analytic theory.

# Acknowledgements

Authors are very grateful to B. Rosenstein, Mingpu Qing, Qiaoyi Li, Qingdong Jiang, Shiping Feng, Tao Wang, Wei Li, Xiaotian Zhang, Xinguo Ren, and Zhipeng Sun for valuable discussions and helps in numerical computations. Authors are deeply grateful to open-source project SmoQyDQMC [22]. This code has been instrumental in applying DQMC results as benchmark within our research, significantly enhancing the efficiency and accuracy of our work.

**Funding information** This work is supported by the National Natural Science Foundation of China (Grant No.12174006 of Prof. Li's fund) and the High-performance Computing Platform of Peking University. H. H. acknowledges the support of the National Key R&D Program of China (No. 2021YFA1401600), the National Natural Science Foundation of China (Grant No. 12474056).

# A    Derivation of Hedin's Equations

Before the derivation, we denote the kinetic term and the interaction term of the action Eq. (4) by $\mathcal{S}_0$ and $\mathcal{S}_I$ for convenience. And we need to couple the external source

$$\mathcal{S}_J[\psi, \psi^*; J] \equiv \sum_a \int d(3) J^a(3) S^a(3) \tag{74}$$

to the system, $\mathcal{S} = \mathcal{S}_0 + \mathcal{S}_I - \mathcal{S}_J$. Then, we start with the field translation invariance,

$$\int \mathcal{D}[\psi, \psi^*] \frac{\delta}{\delta \psi_\alpha^*(1)} \Big[ \psi_\beta^*(2) e^{-\mathcal{S}[\psi, \psi^*]} \Big] = 0. \tag{75}$$

The equation above is equivalent to

$$\delta_{\alpha\beta} \delta(1,2) + \left\langle \psi_\beta^*(2) \frac{\delta \mathcal{S}}{\delta \psi_\alpha^*(1)} \right\rangle = 0. \tag{76}$$

According to the definition of the action Eq. 4,

$$\frac{\delta \mathcal{S}_0}{\delta \psi_\alpha^*(1)} = -\sum_\gamma \int d(3) T_{\alpha\gamma}(1,3) \psi_\gamma(3), \tag{77}$$

$$\frac{\delta \mathcal{S}_I}{\delta \psi_\alpha^*(1)} = -\sum_{ab} \int d(34) \frac{\delta S^a(3)}{\delta \psi_\alpha^*(1)} V^{ab}(3,4) S^b(4). \tag{78}$$

Using the trick described in Sec. 3.1, one has

$$\frac{\delta(\mathcal{S}_0 - \mathcal{S}_J)}{\delta \psi_\alpha^*(1)} = -\sum_\gamma \int d(3) T_{\alpha\gamma}[J](1,3) \psi_\gamma(3), \tag{79}$$

where

$$\boldsymbol{T}[J](1,2) = \boldsymbol{T}(1,2) + \int d(3) J(3) \boldsymbol{K}_{S^a}(1,2;3), \tag{80}$$

$$\boldsymbol{K}_{S^a}(1,2;3) = \boldsymbol{\sigma}^a \delta(1,2)\delta(1,3). \tag{81}$$

The combination of Eqs. (76, 79, 78) gives the Dyson-Schwinger equation

$$
\begin{aligned}
\delta_{\alpha\beta}\delta(1,2) = & \sum_{\gamma} \int d(3)\, T_{\alpha\gamma}[J](1,3) G_{\gamma\beta}(3,2) \\
& + \sum_{ab} \int d(34) V^{ab}(3,4) \left\langle \psi_{\beta}^{*}(2) \frac{\delta S^{a}(3)}{\delta \psi_{\alpha}^{*}(1)} S^{b}(4) \right\rangle.
\end{aligned}
\tag{82}
$$

Making use of the equation

$$\frac{\delta}{\delta J_a(1)} \langle F[\psi,\psi^*] \rangle = \left\langle F[\psi,\psi^*] \frac{\delta \mathcal{S}_J}{\delta J_a(1)} \right\rangle - \langle F[\psi,\psi^*] \rangle \left\langle \frac{\delta \mathcal{S}_J}{\delta J_a(1)} \right\rangle, \tag{83}$$

where $F$ is any functional of $\psi, \psi^*$, and

$$\frac{\delta S^a(3)}{\delta \psi_\alpha^*(1)} = \delta(1,3) \sum_{\gamma} \sigma_{\alpha\gamma}^a \psi_\gamma(3), \tag{84}$$

one obtains that

$$
\begin{aligned}
\left\langle \psi_{\beta}^{*}(2) \frac{\delta S^{a}(3)}{\delta \psi_{\alpha}^{*}(1)} S^{b}(4) \right\rangle = & \, \delta(1,3) \sum_{\gamma} \sigma_{\alpha\gamma}^{a} \frac{\delta G_{\gamma\beta}(3,2)}{\delta J^{b}(4)} \\
& + \delta(1,3) \sum_{\gamma} \sigma_{\alpha\gamma}^{a} G_{\gamma\beta}(3,2) \left\langle S^{b}(4) \right\rangle.
\end{aligned}
\tag{85}
$$

Substituting the Eq. (85) into the Dyson-Schwinger equation, Eq. (82), and then multiplying both sides by the inverse of the Green's function, one has

$$
\begin{aligned}
G_{\alpha\beta}^{-1}(1,2) = & \, T_{\alpha\beta}[J](1,2) + \delta(1,2) \sum_{a} \sigma_{\alpha\beta}^{a} \sum_{b} \int d(4) V^{ab}(1,4) \left\langle S^{b}(4) \right\rangle \\
& + \sum_{ab} \sum_{\mu\nu} \sigma_{\alpha\mu}^{a} \int d(34) V^{ab}(1,3) \frac{\delta G_{\mu\nu}(1,4)}{\delta J^{b}(3)} G_{\nu\beta}^{-1}(4,2).
\end{aligned}
\tag{86}
$$

Notice that $T[J=0] = G_0^{-1}$, thus the self energy is

$$
\begin{aligned}
\Sigma_{\alpha\beta}(1,2) = & \, G_{0\alpha\beta}^{-1}(1,2) - G_{\alpha\beta}^{-1}(1,2) \\[6pt]
= & -\delta(1,2) \sum_{a} \sigma_{\alpha\beta}^{a} \left[ J^{a}(1) + \sum_{b} \int d(4) V^{ab}(1,4) \left\langle S^{b}(4) \right\rangle \right] \\
& - \sum_{ab} \sum_{\mu\nu} \sigma_{\alpha\mu}^{a} \int d(34) V^{ab}(1,3) \frac{\delta G_{\mu\nu}(1,4)}{\delta J^{b}(3)} G_{\nu\beta}^{-1}(4,2),
\end{aligned}
\tag{87}
$$

where the first term of the right hand side is the Hartree self energy

$$\Sigma_{H\alpha\beta}(1,2) = -\delta(1,2) \sum_{a} \sigma_{\alpha\beta}^{a} v^{a}(1) \tag{88}$$

with

$$v^a(1) \equiv J^a(1) + \sum_b \int d(4) V^{ab}(1,4) \langle S^b(4) \rangle, \tag{89}$$

and the rest is denoted by $\Sigma'$ as

$$\Sigma'_{\alpha\beta}(1,2) = -\sum_{ab} \sum_{\mu\nu} \sigma^a_{\alpha\mu} \int d(34) V^{ab}(1,3) \frac{\delta G_{\mu\nu}(1,4)}{\delta J^b(3)} G^{-1}_{\nu\beta}(4,2). \tag{90}$$

Then, define the Hedin vertex $\Gamma_H$ and the functional $W$ as

$$\Gamma^a_{H\alpha\beta}(1,2;3) \equiv \frac{\delta G^{-1}_{\alpha\beta}(1,2)}{\delta v^a(3)}, \tag{91}$$

$$W^{ca}(5,1) \equiv \sum_b \int d(3) \frac{\delta v^c(5)}{\delta J^b(3)} V^{ab}(1,3), \tag{92}$$

and using $-(\delta G/\delta J)G^{-1} = G(\delta G^{-1}/\delta J) = G(\delta G^{-1}/\delta v)(\delta v/\delta J)$, one finally has

$$\Sigma'_{\alpha\beta}(1,2) = \sum_{ac} \sum_{\mu\nu} \sigma^a_{\alpha\mu} \int d(45) G_{\mu\nu}(1,4) \Gamma^c_{H\nu\beta}(4,2;5) W^{ca}(5,1). \tag{93}$$

Applying $\delta/\delta J$ on the Eq. (89), one has

$$\frac{\delta v^c(5)}{\delta J^b(3)} = \delta_{bc}\delta(3,5) + \frac{1}{2} \sum_{de} \sum_{\mu\nu} \sigma^d_{\mu\nu} \int d(46) V^{cd}(5,4) \frac{\delta G_{\nu\mu}(4,4)}{\delta v^e(6)} \frac{\delta v^e(6)}{\delta J^b(3)}. \tag{94}$$

And substituting the equation above into the definition of functional $W$, Eq. (92), one can prove that

$$W^{ca}(5,1) = V^{ca}(5,1) + \sum_{de} \int d(46) V^{cd}(5,4) P^{de}(4,6) W^{ea}(6,1), \tag{95}$$

$$P^{ab}(1,2) = -\frac{1}{2} \sum_{\mu\nu\alpha\beta} \int d(45) \sigma^a_{\mu\nu} G_{\nu\alpha}(1,4) \Gamma^b_{H\alpha\beta}(4,5;2) G_{\beta\mu}(5,1). \tag{96}$$

The Eq. (95) is namely the equation $W^{-1} = V^{-1} - P$. The Eqs. (88, 93, 95, 96) together with $G^{-1} = G_0^{-1} - \Sigma_H - \Sigma'$ form the Hedin Equations. Notice that

$$\begin{aligned}
\Gamma^a_{H\alpha\beta}(1,2;3) &\equiv \frac{\delta G^{-1}_{\alpha\beta}(1,2)}{\delta v^a(3)} = -\frac{\delta \Sigma_{H\alpha\beta}(1,2)}{\delta v^a(3)} - \frac{\delta \Sigma'_{\alpha\beta}(1,2)}{\delta v^a(3)} \\
&= \sigma_{\alpha\beta}\delta(1,2)\delta(1,3) - \frac{\delta \Sigma'_{\alpha\beta}(1,2)}{\delta v^a(3)},
\end{aligned} \tag{97}$$

and the GW approximation ignores the $\delta\Sigma'/\delta v$.

## B  The GW equations and the GW-covariance equations in the momentum representation

On the reciprocal space of the A-B sublattice, the GW equations are

$$[\tilde{G}^{-1}]^{l_1 l_2}(k) = \tilde{T}^{l_1 l_2}(k) - \tilde{\Sigma}_H^{l_1 l_2}(k) - \tilde{\Sigma}_{GW}^{l_1 l_2}(k), \tag{98}$$

$$\tilde{\Sigma}_H^{l_1 l_2}(k) = -\delta_{l_1 l_2} \sum_a \boldsymbol{\sigma}^a \sum_b \sum_{l_3} \tilde{V}^{al_1, bl_3}(0) \, \text{Tr}\left[\boldsymbol{\sigma}^b \boldsymbol{G}^{l_3 l_3}(0)\right], \tag{99}$$

$$\tilde{\Sigma}_{GW}^{l_1 l_2}(k) = \sum_{ab} \frac{1}{\beta N} \sum_q \boldsymbol{\sigma}^a \tilde{G}^{l_1 l_2}(q+k) \boldsymbol{\sigma}^b \tilde{W}^{bl_2, al_1}(q), \tag{100}$$

$$[\tilde{W}^{-1}]^{al_1, bl_2}(q) = [\tilde{V}^{-1}]^{al_1, bl_2}(q) - \tilde{P}^{al_1, bl_2}(q), \tag{101}$$

$$\tilde{P}^{al_1, bl_2}(q) = -\frac{1}{\beta N} \sum_k \text{Tr}\left[\boldsymbol{\sigma}^a \tilde{G}^{l_1 l_2}(k+q) \boldsymbol{\sigma}^b \tilde{G}^{l_2 l_1}(k)\right], \tag{102}$$

where $k = (i\omega_n, \boldsymbol{k})$, $\omega_n$ is the Matsubara frequency, $\omega_n = 2n\pi/\beta$ for boson and $\omega_n = (2n+1)\pi/\beta$ for fermion.

The Fourier transformation of the vertex-like functionals $\boldsymbol{\Lambda}(1-2, 1-3) = \boldsymbol{\Lambda}(1, 2; 3)$ is defined by applying the Eqs. (37, 38) on $1-2$ and $2-3$ respectively. Thus, the correlation function is

$$\tilde{\chi}_{XY}(q) = \frac{1}{\beta N} \sum_k \sum_{l_1 l_2} \text{Tr}\left[\tilde{K}_X^{l_1 l_2}(k, -q) \tilde{\Lambda}_\phi^{l_2 l_1}(k-q, q)\right], \tag{103}$$

and the GW-covariance equations are

$$\tilde{\Gamma}_\phi^{l_1 l_2}(k, q) = \left(\tilde{\gamma}_\phi - \tilde{\Gamma}_\phi^H - \tilde{\Gamma}_\phi^{MT} - \tilde{\Gamma}_\phi^{AL}\right)^{l_1 l_2}(k, q), \tag{104}$$

$$\tilde{\Gamma}_\phi^{H\,l_1 l_2}(k, q) = -\delta_{l_1 l_2} \sum_c \boldsymbol{\sigma}^c \sum_b \sum_{l_4} \tilde{V}^{cl_1, bl_4}(q) \frac{1}{\beta N} \sum_{k'} \text{Tr}\left[\boldsymbol{\sigma}^b \tilde{\Lambda}_\phi^{l_4 l_4}(k', q)\right], \tag{105}$$

$$\tilde{\Gamma}_\phi^{MT\,l_1 l_2}(k, q) = \sum_{cb} \frac{1}{\beta N} \sum_{q'} \boldsymbol{\sigma}^c \tilde{\Lambda}_\phi^{l_1 l_2}(q'+k, q) \boldsymbol{\sigma}^b \tilde{W}^{bl_2, cl_1}(q'), \tag{106}$$

$$\tilde{\Gamma}_\phi^{AL\,l_1 l_2}(k, q) = -\sum_{abcd} \sum_{l_4 l_5} \frac{1}{\beta N} \sum_p \boldsymbol{\sigma}^a \tilde{G}^{l_1 l_2}(k+p+q) \boldsymbol{\sigma}^b$$
$$\times \tilde{W}^{bl_2, cl_4}(p+q) \tilde{\Gamma}_\phi^{W cl_4, dl_5}(p, q) \tilde{W}^{dl_5, al_1}(p), \tag{107}$$

$$\tilde{\Gamma}_\phi^{W dl_4, el_5}(p, q) = \frac{1}{\beta N} \sum_k \text{Tr}\left[\boldsymbol{\sigma}^d \tilde{\Lambda}_\phi^{l_4 l_5}(k+p, q) \boldsymbol{\sigma}^e \tilde{G}^{l_5 l_4}(k)\right.$$
$$\left. + \boldsymbol{\sigma}^d \tilde{G}^{l_4 l_5}(k+p+q) \boldsymbol{\sigma}^e \tilde{\Lambda}_\phi^{l_5 l_4}(k, q)\right], \tag{108}$$

$$\tilde{\Lambda}_\phi^{l_1 l_2}(k, q) = -\sum_{l_4 l_5} \tilde{G}^{l_1 l_4}(k+q) \tilde{\Gamma}_\phi^{l_4 l_5}(k, q) \tilde{G}^{l_5 l_2}(k). \tag{109}$$

## C  Effectiveness of DQMC under Extreme Conditions

We investigate an extreme condition, specifically an extremely low temperature $\beta = 20$ and $U = 8$ in the regime of strong correlations. Within this domain, DQMC results exhibit poor convergence as the system size increases, and the Trotter error arising from the finite $\Delta\tau$ cannot be neglected [4]. To address the challenge of the first difficulty, we focus on a finite-sized system, namely the lattice size is $12 \times 12$. Additionally, we calculate results for various

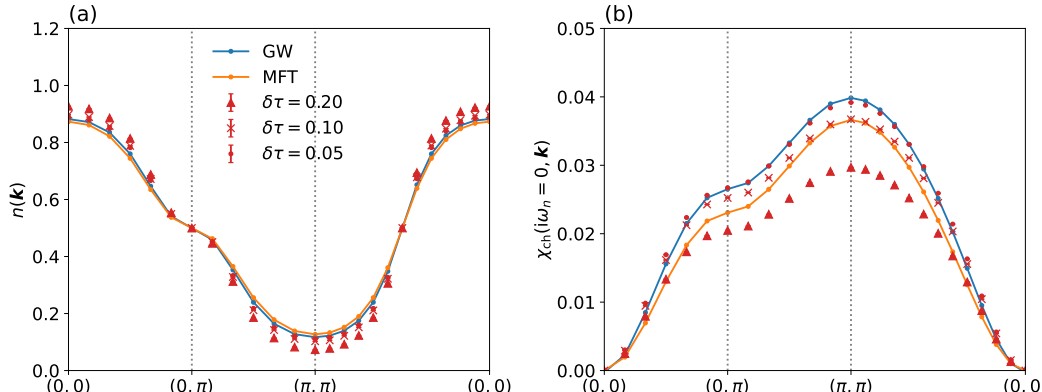

Figure 7: Panels (a) and (b) display the momentum distribution $n(\boldsymbol{k})$ and charge correlation function $\chi_{\text{ch}}$ relatively, which are presented at an extremely low temperatures $\beta = 20$ and a typically strong interaction strength $U = 8$ on a $12 \times 12$ lattice. The results from the GW-covariance and the mean-field-covariance approaches are compared with those from DQMC using different Trotter parameter $\Delta\tau = 0.20, 0.10, 0.05$. The variations over $\Delta\tau$ is neglectable compared with the errorbar.

$\Delta\tau$ values to estimate the magnitude of the Trotter error. Notably, in this extreme condition, the computational cost of many-body methods, exemplified by the GW-covariance and MF-covariance approximation, remains within an acceptable range.

We calculated the momentum distribution $n(\boldsymbol{k})$, which is consistent with that in Ref. [58]

$$n(\boldsymbol{k}) = \frac{1}{N} \sum_{\boldsymbol{r}_1, \boldsymbol{r}_2} \sum_{\sigma} e^{i\boldsymbol{k} \cdot (\boldsymbol{r}_i - \boldsymbol{r}_j)} \langle \hat{c}_{\sigma}^{\dagger}(\boldsymbol{r}_i) \hat{c}_{\sigma}(\boldsymbol{r}_j) \rangle, \tag{110}$$

and the charge correlations $\chi_{\text{ch}} \equiv \chi_{S^0 S^0}$ (recalling Eq. 11) at $i\omega_n = 0$ in the momentum space. These two represent single-particle properties and two-particle properties, respectively. As shown in Fig. 7, $\Delta\tau = 0.1$ for the momentum distribution $n(\boldsymbol{k})$ and $\Delta\tau = 0.05$ for the charge correlation function $\chi_{\text{ch}}(i\omega_n = 0, \boldsymbol{k})$ are small enough to make the DQMC simulations converge. When comparing the $\chi_{\text{ch}}$ calculated by GW-covariance, MF-covariance and DQMC with $\Delta\tau = 0.1$, one will mistakenly believe that the MF-covariance exhibits better behavior than the GW-covariance, although the fact shown by DQMC with $\Delta\tau = 0.05$ is that the GW-covariance perfactly matches the DQMC. Thus, it is necessary to ensure that the DQMC data used for benchmarking converges.

# D    Results at half filling with intermediate coupling $U = 4$

We also investage the intermediate coupling $U = 4$ system at half filling on $16 \times 16$ lattice, results are shown in Fig. 8, Fig. 9 and Fig. 10 relatively. The $U = 4$ results show very similar properties to Fig. 2, Fig. 3 and Fig. 5. For example, at low temperature away from the pseudo critical temperature $\beta_c \simeq 0.22$, the GW agree with the corresponding DQMC benchmark. This further indicates that the effectiveness of the symmetrization scheme in the strong coupling $U = 8$ system is not accidental.

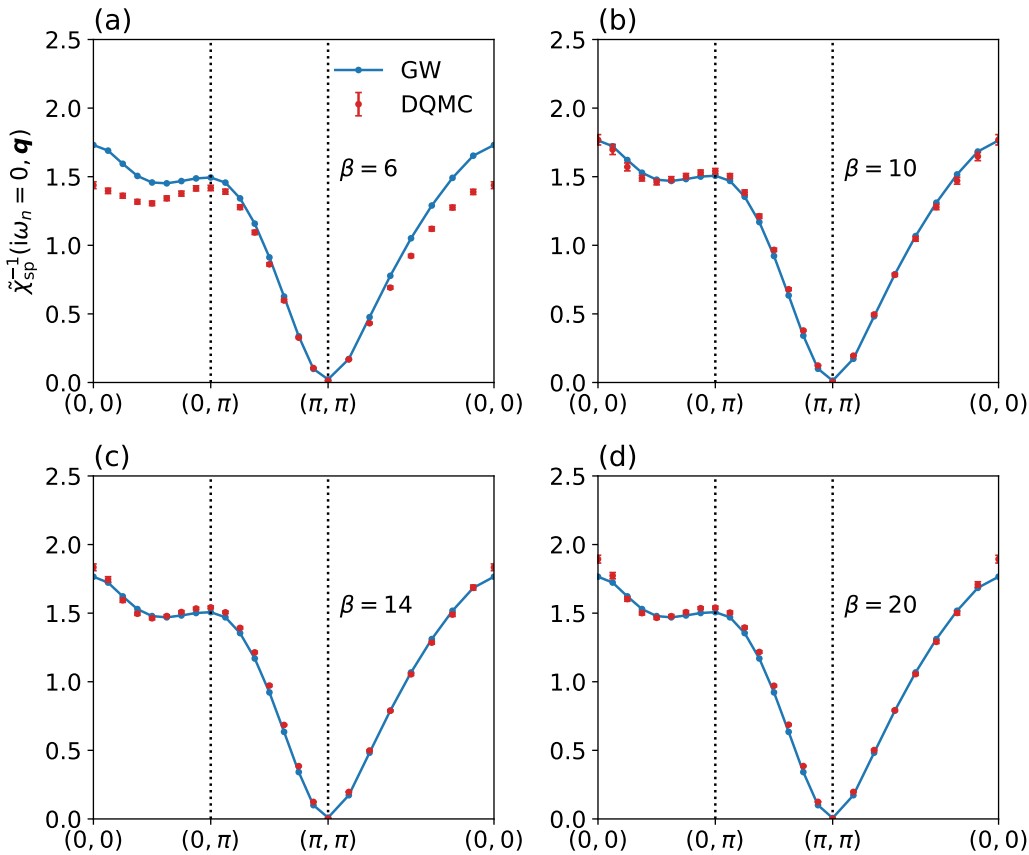

Figure 8: Momentum dependence of the inverse spin susceptibility for the half-filled Hubbard model on a $16 \times 16$ lattice with $U = 4$. Results are shown for decreasing temperatures, corresponding to inverse temperatures (a) $\beta = 6$, (b) $\beta = 10$, (c) $\beta = 14$, and (d) $\beta = 20$. The momentum path follows $(0, 0) \rightarrow (0, \pi) \rightarrow (\pi, \pi) \rightarrow (0, 0)$. Blue curves denote the symmetrized GW-covariance approximation, while red symbols with error bars show DQMC benchmarks. A systematic improvement in the agreement between GW-covariance and DQMC is observed as temperature is lowered.

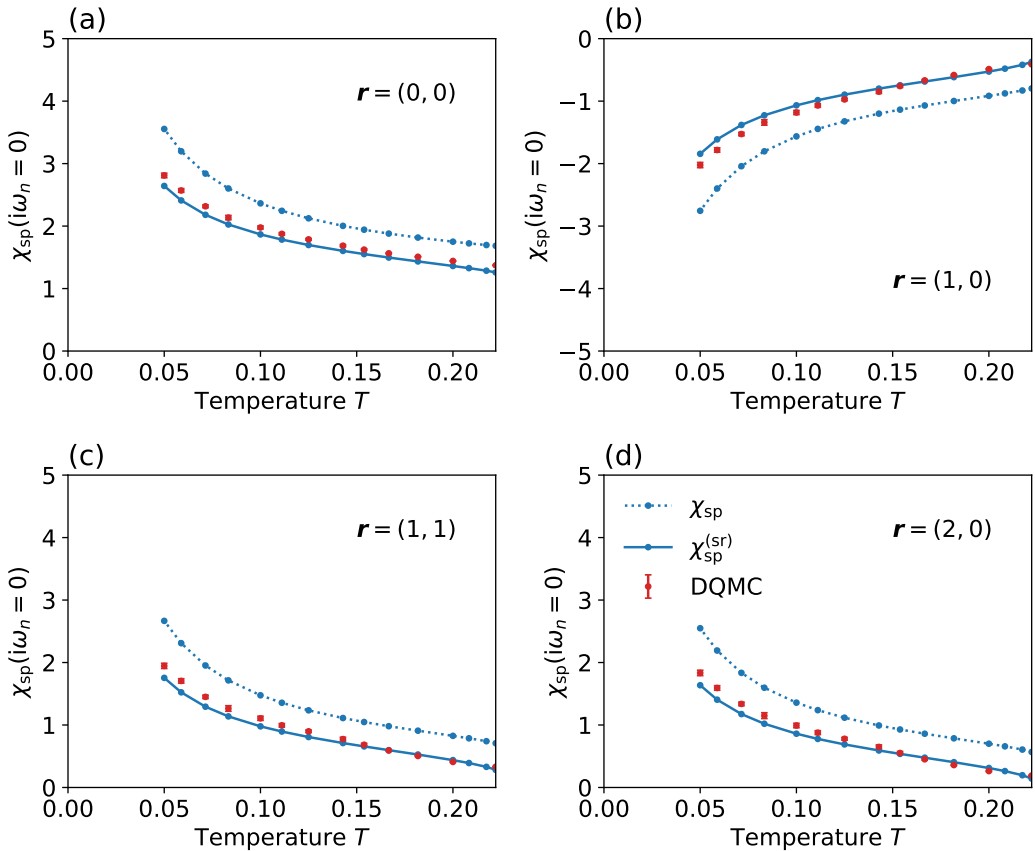

Figure 9: Temperature dependence of the static spin correlation function in spatial space for the half-filled Hubbard model ($U = 4$) on a $16 \times 16$ lattice. Panels (a)-(d) display results for lattice separations $\mathbf{r} = (0,0)$, $(1,0)$, $(1,1)$, and $(2,0)$, respectively. The DQMC benchmarks (red points with error bars) are compared with the GW-covariance result $\chi_{\mathrm{sp}}$ (blue dashed curve) and the $\chi$-sum rule estimate $\chi_{\mathrm{sp}}^{(\mathrm{sr})}$ (blue solid curve).

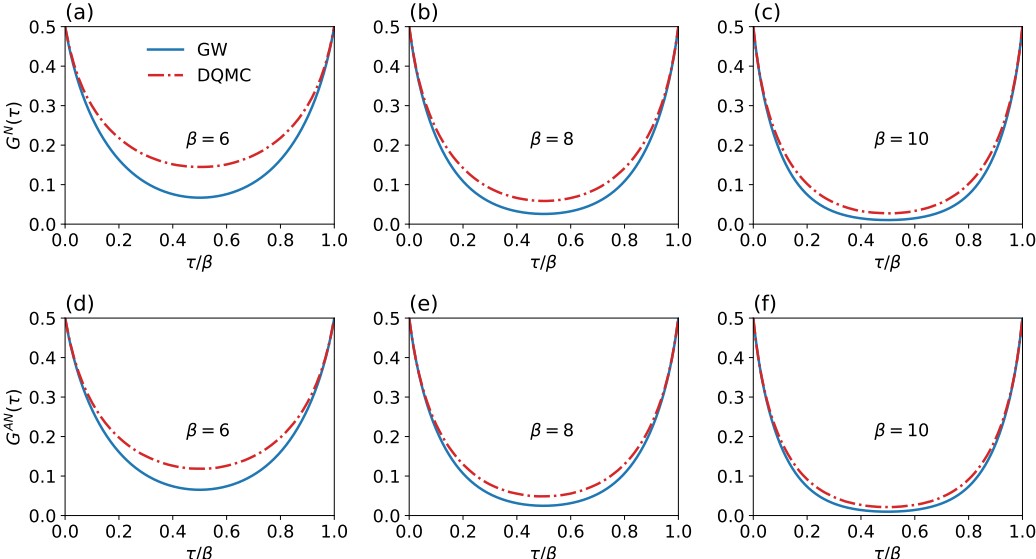

Figure 10: Imaginary-time Green's function $G(\mathbf{k}, \tau)$ for the half-filled Hubbard model on a $16 \times 16$ lattice with $U = 4$, comparing the symmetrized GW approximation (blue solid lines) and DQMC results (red dot-dashed lines). Panels (a)-(c) correspond to the nodal point $\mathbf{k} = (\pi/2, \pi/2)$, while panels (d)-(f) show the antinodal point $\mathbf{k} = (\pi, 0)$. The columns represent increasing inverse temperatures, i.e., (a,d) $\beta = 6$, (b,e) $\beta = 8$, and (c,f) $\beta = 10$.

# E On the $\chi$-sum rule and Pauli exclusion principle

In the section, we present the derivation of the $\chi$-sum rule provided in Ref. [16] and discuss it. Here we denote the spin-resolved density operator as $\hat{n}_\alpha(1)$ with $\alpha = \uparrow, \downarrow$ for clarity. The charge density operator is $\hat{n}(1) = \hat{n}_\uparrow(1) + \hat{n}_\downarrow(1)$, and the spin-z density operator is $\hat{S}^z(1) = \hat{n}_\uparrow(1) - \hat{n}_\downarrow(1)$. Using this notation, the onsite charge function at $\tau = 0$ becomes

$$
\begin{aligned}
\chi_{\mathrm{ch}}(1,1) &\equiv \langle \hat{n}(1)\hat{n}(1) \rangle_{\mathrm{C}} = \langle \hat{n}(1)\hat{n}(1) \rangle - \rho^2 \\
&= \langle \hat{n}_\uparrow(1)\hat{n}_\uparrow(1) \rangle + \langle \hat{n}_\downarrow(1)\hat{n}_\downarrow(1) \rangle + 2\langle \hat{n}_\uparrow(1)\hat{n}_\downarrow(1) \rangle - \rho^2,
\end{aligned}
\tag{111}
$$

where $\rho = \langle \hat{n} \rangle$ is the charge density. The spin correction becomes

$$
\begin{aligned}
\chi_{\mathrm{sp}}(1,1) &\equiv \langle \hat{S}^z(1)\hat{S}^z(1) \rangle \\
&= \langle \hat{n}_\uparrow(1)\hat{n}_\uparrow(1) \rangle + \langle \hat{n}_\downarrow(1)\hat{n}_\downarrow(1) \rangle - 2\langle \hat{n}_\uparrow(1)\hat{n}_\downarrow(1) \rangle.
\end{aligned}
\tag{112}
$$

Combining these two equations, one has

$$
\chi_{\mathrm{ch}}(1,1) + \chi_{\mathrm{sp}}(1,1) = 2\langle \hat{n}_\uparrow(1)\hat{n}_\uparrow(1) \rangle + 2\langle \hat{n}_\downarrow(1)\hat{n}_\downarrow(1) \rangle - \rho^2.
\tag{113}
$$

The Pauli exclusion principle indicates that the eigenvalues of a fermion density operator $\hat{n}_\alpha(1)$ are either 0 or 1. As a results, $\langle \hat{n}_\alpha(1)\hat{n}_\alpha(1) \rangle = \langle \hat{n}_\alpha(1) \rangle$, thus

$$
\chi_{\mathrm{ch}}(\tau = 0, \boldsymbol{r} = \boldsymbol{0}) + \chi_{\mathrm{sp}}(\tau = 0, \boldsymbol{r} = \boldsymbol{0}) = 2\rho - \rho^2.
\tag{114}
$$

Eq. (114) is commonly known as the $\chi$-sum rule. The left hand side of Eq. (114) are two particle properties $\chi_{\mathrm{ch}}$ and $\chi_{\mathrm{sp}}$, while the right hand side is the charge density $\rho$, which a single particle property.

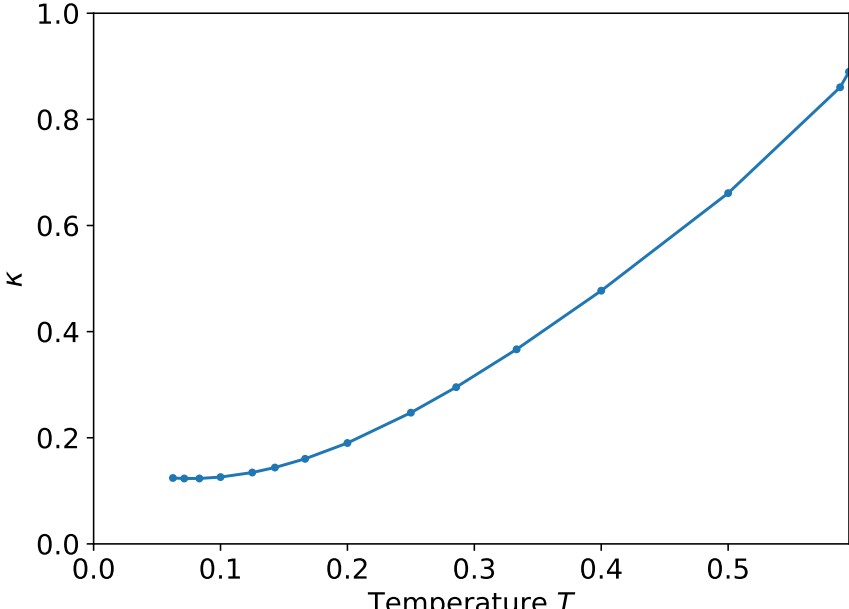

Figure 11: Deviation of $\chi$-sum rule, quantitatively judged by $\kappa$ as defined in Eq. (115), of the pseudo AF solutions calculated by GW-covariance approach in $U = 8$ half-filling Hubbard model on the $12 \times 12$ lattice.

It is meaningful to examine the degree to which our approach violates the Pauli exclusion principle. However, we cannot directly quantify this property. As a proxy, we investigate the deviation from the $\chi$-sum rule by introducing

$$\kappa \equiv \left| \frac{\chi_{\text{ch}}(\tau = 0, \boldsymbol{r} = \boldsymbol{0}) + \chi_{\text{sp}}(\tau = 0, \boldsymbol{r} = \boldsymbol{0})}{2\rho - \rho^2} - 1 \right|, \tag{115}$$

which is the relative error between two- and single-particle properties in Eq. (114). The $\chi$-sum rule is one representation of the local momentum sum rules based on the Pauli exclusion principle, and thus is a necessary condition for a system to obey this principle. We calculate $\kappa$, which reflects the deviations from the $\chi$-sum rule, for a strongly coupled system ($U = 8$) at half-filling on a $12 \times 12$ lattice. The results are presented in Fig. 11. As shown in Fig. 11, $\kappa$ decreases from $\sim 90\%$ to $\sim 10\%$ as the temperature drops below the pseudo critical temperature $T_{\text{c}}$. This trend in $\kappa$ appears to correlate with the reliability of the results, supporting our conjecture in Sec. 5. In the region near $T_{\text{c}}$ ($0.3 < T < T_{\text{c}}$), where the $\chi$-sum rule violation is large ($\kappa$ is high), the results of the symmetrized GW-covariance approximation (shown in Fig. 3) are less reliable. Conversely, at the deep low temperatures far from $T_{\text{c}}$, where the $\chi$-sum rule is better satisfied ($\kappa$ is small), the symmetrized GW-covariance results exhibit better agreement with the DQMC benchmarks.

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
