# Peer review of "Nonpertubative Many-Body Theory for the Two-Dimensional Hubbard Model at Low Temperature: From Weak to Strong Coupling Regimes"

_SciPost Physics_

## Round 1 · Referee Report · Anonymous (Referee 1) · 2025-12-4

Strengths

1- the paper demonstrates a computational approach that seems to work. The method is benchmarked with the numerically exact DQMC method: An excellent agreement was found for a wide range of parameter values.

2- the work is very well motivated and connected to previous works

3- ample details of the formalism are given and the method is well explained

4- the proposed method is numerically inexpensive and easy to implement

5- having the above in mind, I would say that there is strong potential for follow-up works, and people might be interested in trying out the method themselves.

Weaknesses

1- the text is very long, and not well organized. It is difficult to extract the main points. I would say at least 20-30% of the text is either irrelevant or not directly related to the main line of presentation and could be moved to the appendix. There are many parts which read like a review paper, and hand-waving explanations of well known concepts which can only cause confusion. There is also some redundancy, and some points are discussed in several separate places in the manuscript, which makes it hard to navigate the paper.

2- the Fierz ambiguity is not discussed. The Eq.31 is not the only way to transform the Hamiltonian in preparation for the GW treatment. There is an arbitrariness in the way GW is formulated, in the sense that one can tune the decoupling scheme between the charge, spin and pairing channels. This point is somewhat important and should not be entirely overlooked.

3- the "$\chi$-sum rule estimate" of the static susceptibilities shown in Fig.3,4 and 9 ($\chi_\mathrm{sp}^{(sr)}$) is not well explained

4- the method is not discussed in the context of the paradigm of controlled approximations: How can we be sure that the method will perform well outside of the parameter range where it was tested? Does it become exact in any limits? Can one formulate a series of systematic improvements to lead the method towards the exact solution?

5- $\chi_\mathrm{sp}^{(sr)}$ in Fig.3 in some cases apparently crosses zero. This potentially unphysical result is not sufficiently discussed.

Report

The authors formulate a symmetrization scheme within the covariance-GW method that allows one to push this method to lower temperature and stronger couplings. GW calculations in 2D are known to run into divergence of the auxiliary boson in the spin-channel, which prevents calculations at strong coupling and low temperatures. The divergence of the spin-boson is related to the emergence of an artificial antiferromagnetic (AF) order, where one would not expect it, due to the Mermin-Wagner theorem. The idea of the paper is that an AF broken-symmetry calculation can be performed, and that the unordered (physically relevant) results can then be recovered by averaging over all the possible orientations of the order parameter (i.e. spin averages). To my understanding (I feel this point is not sufficiently well explained in the text), the static susceptibilities at momentum $Q$ are computed in a post-processing step that enforces the chi-sum, based on the single-particle properties encoded in the Green's function (the electron density) and the computed dynamical susceptibilites (at all the other $q$-vectors and non-zero frequencies). With this post-processing step, the agreement with the reference method (DQMC) is found to be good, but otherwise there is quite some discrepancy. The appeal of the method is that it provides a cheap way towards reasonable (approximate) computations at strong coupling and low temperatures. Previously, the divergence of the spin-boson hindered precise estimates of the superconducting $T_c$ in the Hubbard model at the level of GW theory, and the present scheme might provide a way forward in such cases.

The paper presents an interesting idea and the results appear very convincing. I believe that the paper easily satisfies the acceptance criterion:

"Open a new pathway in an existing or a new research direction, with clear potential for multi-pronged follow-up work"

The paper presents a possible way forward in GW methodology which is otherwise very widely accepted and often applied in many contexts, and for a good reason.

I think that the explanations of the method and the the presentation of the results are good for the most part. All the important details are covered and one could easily try to reimplement the method and reproduce the results. However, in many places there are even too many unnecessary details and digressions that draw attention away from the main line of presentation. For this reason, the paper is hard to navigate, which reduces readability and may lead to misunderstanding on the part of the reader.

In general, I find that the paper is excellent, but my impression is that the presentation needs to be improved and that a couple of more clarifications and comments need to be added before the paper fully satisfies the criterion:

"Be written in a clear and intelligible way, free of unnecessary jargon, ambiguities and misrepresentations"

Requested changes

1- shorten and reorganize the text to make it more readable and the main points more easily accessible. The authors can decide how exactly to achieve that, but I suggest the following:

a) broad review of literature could be moved to Appendix or the Discussion section, while the motivation and the context in the introductory sections are kept short and to-the-point. b) the part of the text explaining the method (what is being computed and how it is computed) should be well separated from the discussions of other aspects of the work, e.g. context, motivation, good scientific practice tenets etc.
c) each point is discussed in a single place if possible

2- expand the explanations around the $\chi$-sum rule estimation of $\chi^{(sr)}$, Eq.72-73. Does this effectively impose the $\chi$-sum rule? If this approach works, does it mean that the covariance-GW result for $\chi(i\omega_n)$ is good for all frequencies except for $i\omega_n=0$? How exactly is $\chi_\mathbf{r}$ obtained (shown in Fig.3)? Is the calculation first done in $q$-space, and then the result is Fourier transformed to the real-space? Or is Eq.72 written in $i\omega_n$,$\mathbf{r}$-space used directly, analogously to Eq.73?

3- discuss Fierz ambiguity around Eq.31

4- discuss the method in terms of the controlled approximations paradigm.

5- discuss more fairly any anomalous or unphysical properties of the theory

Recommendation

Ask for minor revision

---

## Round 1 · Referee Report · Anonymous (Referee 2) · 2025-12-8

Strengths

1-important topic and motivation 2- good benchmarking and truthful evaluation of new methodology 3-should motivate further work in the direction

Weaknesses

1- regions of numerical validity are not clearly defined or knowable 2-lacking discussion on further methods of improvement or connection to controlled approximation schemes

Report

I’ve reviewed the work by Xiao and others. They present a symmetrization to the GW-covariance approach and make comparisons to benchmark DQMC calculations for the half-filled hubbard model, primarily concerned with temperature and U dependence.

There is a long history of a largely unsolved problem that the authors are trying to tackle, that being finite temperature phase transitions in 2D systems, or rather that there is violation of mermin wagner theorem when evaluating two particle susceptibilities that leads to divergences and false transitions at some particular temperature and beyond some critical U value. The primary result of the present work seems to be the ability to evaluate the problem on either side of the transition and this is accomplished by making sure that the approach satisfies the ward identity.

Overall, I think the manuscript is a meaningful, though technical, advancement that deserves further attention and scrutiny. I don’t think that the authors have convinced me that this methodology is the best way forward, but I do think that the work is sufficiently detailed and rigorous to justify publication.

The only concern I would like to raise is regarding a note in the captions of figures 2 and 8. The authors state that the data improves as temperature is lowered. However, I do not visually see that this is the case, and in some other datasets it seems that often higher temperatures do quite well with the approach. Is there a general statement here that is always true? Unless the approach is exact in the T=0 limit it isn’t obvious to me that the data improves as temperature is lowered. I think the authors should revise that statement.

I have no further comments to improve the paper, it can be published after considering these revisions.

Requested changes

1-revisit the statements about improved results on lowering temperature. Either quantify this statement clearly or remove.

Recommendation

Publish (easily meets expectations and criteria for this Journal; among top 50%)

---

## Editorial Decision

awaiting_resubmission